# On the Information Bottleneck
# Theory of Deep Learning

**Andrew M. Saxe, Yamini Bansal, Joel Dapello, Madhu Advani**
Harvard University
`{asaxe,madvani}@fas.harvard.edu,{ybansal,dapello}@g.harvard.edu`

**Artemy Kolchinsky, Brendan D. Tracey**
Santa Fe Institute
`{artemyk,tracey.brendan}@gmail.com`

**David D. Cox**
Harvard University
MIT-IBM Watson AI Lab
`davidcox@fas.harvard.edu`
`david.d.cox@ibm.com`

## Abstract

The practical successes of deep neural networks have not been matched by theoretical progress that satisfyingly explains their behavior. In this work, we study the information bottleneck (IB) theory of deep learning, which makes three specific claims: first, that deep networks undergo two distinct phases consisting of an initial fitting phase and a subsequent compression phase; second, that the compression phase is causally related to the excellent generalization performance of deep networks; and third, that the compression phase occurs due to the diffusion-like behavior of stochastic gradient descent. Here we show that none of these claims hold true in the general case. Through a combination of analytical results and simulation, we demonstrate that the information plane trajectory is predominantly a function of the neural nonlinearity employed: double-sided saturating nonlinearities like tanh yield a compression phase as neural activations enter the saturation regime, but linear activation functions and single-sided saturating nonlinearities like the widely used ReLU in fact do not. Moreover, we find that there is no evident causal connection between compression and generalization: networks that do not compress are still capable of generalization, and vice versa. Next, we show that the compression phase, when it exists, does not arise from stochasticity in training by demonstrating that we can replicate the IB findings using full batch gradient descent rather than stochastic gradient descent. Finally, we show that when an input domain consists of a subset of task-relevant and task-irrelevant information, hidden representations do compress the task-irrelevant information, although the overall information about the input may monotonically increase with training time, and that this compression happens concurrently with the fitting process rather than during a subsequent compression period.

## 1 Introduction

Deep neural networks (Schmidhuber, 2015; LeCun et al., 2015) are the tool of choice for real-world tasks ranging from visual object recognition (Krizhevsky et al., 2012), to unsupervised learning (Goodfellow et al., 2014; Lotter et al., 2016) and reinforcement learning (Silver et al., 2016). These practical successes have spawned many attempts to explain the performance of deep learning systems (Kadmon & Sompolinsky, 2016), mostly in terms of the properties and dynamics of the optimization problem in the space of weights (Saxe et al., 2014; Choromanska et al., 2015; Advani & Saxe, 2017), or the classes of functions that can be efficiently represented by deep networks (Montufar et al., 2014; Poggio et al., 2017). This paper analyzes a recent inventive proposal to study the dynamics of learning through the lens of information theory (Tishby & Zaslavsky, 2015; Shwartz-Ziv & Tishby, 2017). In this view, deep learning is a question of representation learning: each layer of a deep neural network can be seen as a set of summary statistics which contain some but not all of the information present in the input, while retaining as much information about the target output as possible. The

amount of information in a hidden layer regarding the input and output can then be measured over the course of learning, yielding a picture of the optimization process in the information plane. Crucially, this method holds the promise to serve as a general analysis that can be used to compare different architectures, using the common currency of mutual information. Moreover, the elegant information bottleneck (IB) theory provides a fundamental bound on the amount of input compression and target output information that any representation can achieve (Tishby et al., 1999). The IB bound thus serves as a method-agnostic ideal to which different architectures and algorithms may be compared.

A preliminary empirical exploration of these ideas in deep neural networks has yielded striking findings (Shwartz-Ziv & Tishby, 2017). Most saliently, trajectories in the information plane appear to consist of two distinct phases: an initial "fitting" phase where mutual information between the hidden layers and both the input and output increases, and a subsequent "compression" phase where mutual information between the hidden layers and the input *decreases*. It has been hypothesized that this compression phase is responsible for the excellent generalization performance of deep networks, and further, that this compression phase occurs due to the random diffusion-like behavior of stochastic gradient descent.

Here we study these phenomena using a combination of analytical methods and simulation. In Section 2, we show that the compression observed by Shwartz-Ziv & Tishby (2017) arises primarily due to the double-saturating tanh activation function used. Using simple models, we elucidate the effect of neural nonlinearity on the compression phase. Importantly, we demonstrate that the ReLU activation function, often the nonlinearity of choice in practice, does not exhibit a compression phase. We discuss how this compression via nonlinearity is related to the assumption of binning or noise in the hidden layer representation. To better understand the dynamics of learning in the information plane, in Section 3 we study deep linear networks in a tractable setting where the mutual information can be calculated exactly. We find that deep linear networks do not compress over the course of training for the setting we examine. Further, we show a dissociation between generalization and compression. In Section 4, we investigate whether stochasticity in the training process causes compression in the information plane. We train networks with full batch gradient descent, and compare the results to those obtained with stochastic gradient descent. We find comparable compression in both cases, indicating that the stochasticity of SGD is not a primary factor in the observed compression phase. Moreover, we show that the two phases of SGD occur even in networks that do not compress, demonstrating that the phases are not causally related to compression. These results may seem difficult to reconcile with the intuition that compression can be necessary to attain good performance: if some input channels primarily convey noise, good generalization requires excluding them. Therefore, in Section 5 we study a situation with explicitly task-relevant and task-irrelevant input dimensions. We show that the hidden-layer mutual information with the task-irrelevant subspace does indeed drop during training, though the overall information with the input increases. However, instead of a secondary compression phase, this task-irrelevant information is compressed at the same time that the task-relevant information is boosted. Our results highlight the importance of noise assumptions in applying information theoretic analyses to deep learning systems, and put in doubt the generality of the IB theory of deep learning as an explanation of generalization performance in deep architectures.

## 2 COMPRESSION AND NEURAL NONLINEARITIES

The starting point for our analysis is the observation that changing the activation function can markedly change the trajectory of a network in the information plane. In Figure 1A, we show our replication of the result reported by Shwartz-Ziv & Tishby (2017) for networks with the $\tanh$ nonlinearity.[1] This replication was performed with the code supplied by the authors of Shwartz-Ziv & Tishby (2017), and closely follows the experimental setup described therein. Briefly, a neural network with 7 fully connected hidden layers of width 12-10-7-5-4-3-2 is trained with stochastic gradient descent to produce a binary classification from a 12-dimensional input. In our replication we used 256 randomly selected samples per batch. The mutual information of the network layers with respect to the input and output variables is calculated by binning the neuron's $\tanh$ output activations into 30 equal intervals between -1 and 1. Discretized values for each neuron in each layer are then used to directly calculate the joint distributions, over the 4096 equally likely input patterns and true output labels. In line with prior work (Shwartz-Ziv & Tishby, 2017), the dynamics in Fig. 1 show a

---

[1]Code for our results is available at `https://github.com/artemyk/ibsgd/tree/iclr2018`

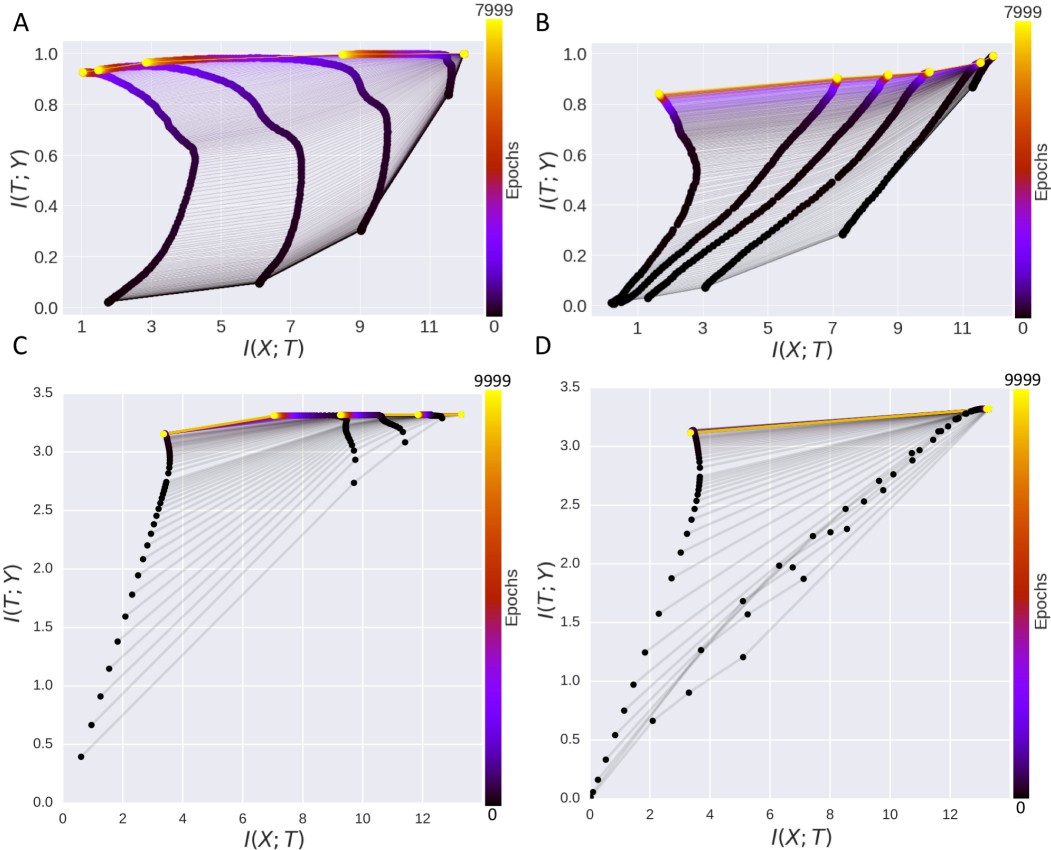

Figure 1: Information plane dynamics and neural nonlinearities. (A) Replication of Shwartz-Ziv & Tishby (2017) for a network with $\mathrm{tanh}$ nonlinearities (except for the final classification layer which contains two sigmoidal neurons). The x-axis plots information between each layer and the input, while the y-axis plots information between each layer and the output. The color scale indicates training time in epochs. Each of the six layers produces a curve in the information plane with the input layer at far right, output layer at the far left. Different layers at the same epoch are connected by fine lines. (B) Information plane dynamics with ReLU nonlinearities (except for the final layer of 2 sigmoidal neurons). Here no compression phase is visible in the ReLU layers. For learning curves of both networks, see Appendix A. (C) Information plane dynamics for a $\mathrm{tanh}$ network of size $784 - 1024 - 20 - 20 - 20 - 10$ trained on MNIST, estimated using the non-parametric kernel density mutual information estimator of Kolchinsky & Tracey (2017); Kolchinsky et al. (2017), no compression is observed except in the final classification layer with sigmoidal neurons. See Appendix B for the KDE MI method applied to the original Tishby dataset; additional results using a second popular nonparametric k-NN-based method (Kraskov et al., 2004); and results for other neural nonlinearities.

transition between an initial fitting phase, during which information about the input increases, and a subsequent compression phase, during which information about the input decreases.

We then modified the code to train deep networks using rectified linear activation functions ($f(x) = \max(0, x)$). While the activities of $\tanh$ networks are bounded in the range $[-1, 1]$, ReLU networks have potentially unbounded positive activities. To calculate mutual information, we first trained the ReLU networks, next identified their largest activity value over the course of training, and finally chose 100 evenly spaced bins between the minimum and maximum activity values to discretize the hidden layer activity. The resulting information plane dynamics are shown in Fig. 1B. The mutual information with the input monotonically increases in all ReLU layers, with no apparent compression phase. To see whether our results were an artifact of the small network size, toy dataset, or simple binning-based mutual information estimator we employed, we also trained larger networks on the MNIST dataset and computed mutual information using a state-of-the-art nonparametric kernel density estimator which assumes hidden activity is distributed as a mixture of Gaussians (see Appendix B for details). Fig. C-D show that, again, $\tanh$ networks compressed but ReLU networks did not. Appendix B shows that similar results also obtain with the popular nonparametric k-nearest-neighbor estimator of Kraskov et al. (2004), and for other neural nonlinearities. Thus, the choice of nonlinearity substantively affects the dynamics in the information plane.

To understand the impact of neural nonlinearity on the mutual information dynamics, we develop a minimal model that exhibits this phenomenon. In particular, consider the simple three neuron network shown in Fig. 2A. We assume a scalar Gaussian input distribution $X \sim \mathcal{N}(0, 1)$, which is fed through the scalar first layer weight $w_1$, and passed through a neural nonlinearity $f(\cdot)$, yielding the hidden unit activity $h = f(w_1 X)$. To calculate the mutual information with the input, this hidden unit activity is then binned yielding the new discrete variable $T = \text{bin}(h)$ (for instance, into 30 evenly spaced bins from -1 to 1 for the $\tanh$ nonlinearity). This binning process is depicted in Fig. 2B. In this simple setting, the mutual information $I(T; X)$ between the binned hidden layer activity $T$ and the input $X$ can be calculated exactly. In particular,

$$
\begin{aligned}
I(T; X) &= H(T) - H(T|X) & (1) \\
&= H(T) & (2) \\
&= -\sum_{i=1}^{N} p_i \log p_i & (3)
\end{aligned}
$$

where $H(\cdot)$ denotes entropy, and we have used the fact that $H(T|X) = 0$ since $T$ is a deterministic function of $X$. Here the probabilities $p_i = P(h \geq b_i \text{ and } h < b_{i+1})$ are simply the probability that an input $X$ produces a hidden unit activity that lands in bin $i$, defined by lower and upper bin limits $b_i$ and $b_{i+1}$ respectively. This probability can be calculated exactly for monotonic nonlinearities $f(\cdot)$ using the cumulative density of $X$,

$$
p_i = P(X \geq f^{-1}(b_i)/w_1 \text{ and } X < f^{-1}(b_{i+1})/w_1), \quad (4)
$$

where $f^{-1}(\cdot)$ is the inverse function of $f(\cdot)$.

As shown in Fig. 2C-D, as a function of the weight $w_1$, mutual information with the input first increases and then decreases for the $\tanh$ nonlinearity, but always increases for the ReLU nonlinearity. Intuitively, for small weights $w_1 \approx 0$, neural activities lie near zero on the approximately linear part of the $\tanh$ function. Therefore $f(w_1 X) \approx w_1 X$, yielding a rescaled Gaussian with information that grows with the size of the weights. However for very large weights $w_1 \to \infty$, the $\tanh$ hidden unit nearly always saturates, yielding a discrete variable that concentrates in just two bins. This is more or less a coin flip, containing mutual information with the input of approximately 1 bit. Hence the distribution of $T$ collapses to a much lower entropy distribution, yielding compression for large weight values. With the ReLU nonlinearity, half of the inputs are negative and land in the bin containing a hidden activity of zero. The other half are Gaussian distributed, and thus have entropy that increases with the size of the weight.

Hence double-saturating nonlinearities can lead to compression of information about the input, as hidden units enter their saturation regime, due to the binning procedure used to calculate mutual information. The crux of the issue is that the actual $I(h; X)$ is infinite, unless the network itself adds noise to the hidden layers. In particular, without added noise, the transformation from $X$ to the continuous hidden activity $h$ is deterministic and the mutual information $I(h; X)$ would generally be

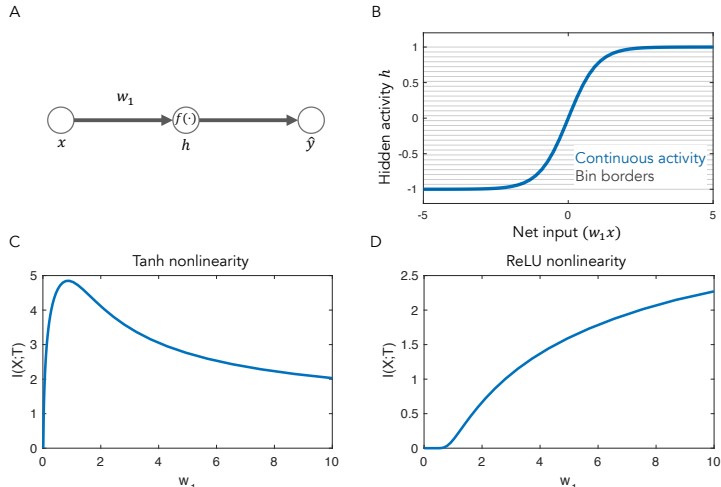

Figure 2: Nonlinear compression in a minimal model. (A) A three neuron nonlinear network which receives Gaussian inputs $x$, multiplies by weight $w_1$, and maps through neural nonlinearity $f(\cdot)$ to produce hidden unit activity $h$. (B) The continuous activity $h$ is binned into a discrete variable $T$ for the purpose of calculating mutual information. Blue: continuous $\tanh$ nonlinear activation function. Grey: Bin borders for 30 bins evenly spaced between -1 and 1. Because of the saturation in the sigmoid, a wide range of large magnitude net input values map to the same bin. (C) Mutual information with the input as a function of weight size $w_1$ for a $\tanh$ nonlinearity. Information increases for small $w_1$ and then decreases for large $w_1$ as all inputs land in one of the two bins corresponding to the saturation regions. (D) Mutual information with the input for the ReLU nonlinearity increases without bound. Half of all inputs land in the bin corresponding to zero activity, while the other half have information that scales with the size of the weights.

infinite (see Appendix C for extended discussion). Networks that include noise in their processing (e.g., Kolchinsky et al. (2017)) can have finite $I(T; X)$. Otherwise, to obtain a finite MI, one must compute mutual information as though there were binning or added noise in the activations. But this binning/noise is not actually a part of the operation of the network, and is therefore somewhat arbitrary (different binning schemes can result in different mutual information with the input, as shown in Fig. 14 of Appendix C).

We note that the binning procedure can be viewed as implicitly adding noise to the hidden layer activity: a range of $X$ values map to a single bin, such that the mapping between $X$ and $T$ is no longer perfectly invertible (Laughlin, 1981). The binning procedure is therefore crucial to obtaining a finite MI value, and corresponds approximately to a model where noise enters the system after the calculation of $h$, that is, $T = h + \epsilon$, where $\epsilon$ is noise of fixed variance independent from $h$ and $X$. This approach is common in information theoretic analyses of deterministic systems, and can serve as a measure of the complexity of a system's representation (see Sec 2.4 of Shwartz-Ziv & Tishby (2017)). However, neither binning nor noise is present in the networks that Shwartz-Ziv & Tishby (2017) considered, nor the ones in Fig. 2, either during training or testing. It therefore remains unclear whether robustness of a representation to this sort of noise in fact influences generalization performance in deep learning systems.

Furthermore, the addition of noise means that different architectures may no longer be compared in a common currency of mutual information: the binning/noise structure is arbitrary, and architectures that implement an identical input-output map can nevertheless have different robustness to noise added in their internal representation. For instance, Appendix C describes a family of linear networks that compute exactly the same input-output map and therefore generalize identically, but yield different mutual information with respect to the input. Finally, we note that approaches which view the weights obtained from the training process as the random variables of interest may sidestep this issue (Achille & Soatto, 2017).

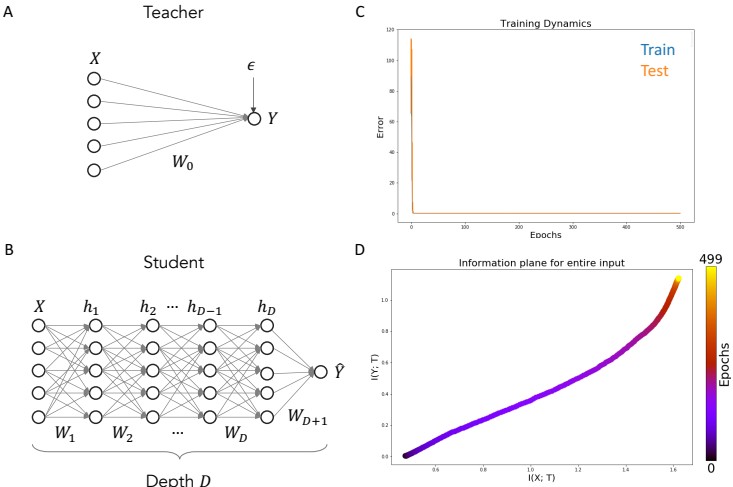

Figure 3: Generalization and information plane dynamics in deep linear networks. (A) A linear teacher network generates a dataset by passing Gaussian inputs $X$ through its weights and adding noise. (B) A deep linear student network is trained on the dataset (here the network has 1 hidden layer to allow comparison with Fig. 4A, see Supplementary Figure 18 for a deeper network). (C) Training and testing error over time. (D) Information plane dynamics. No compression is observed.

Hence when a `tanh` network is initialized with small weights and over the course of training comes to saturate its nonlinear units (as it must to compute most functions of practical interest, see discussion in Appendix D), it will enter a compression period where mutual information decreases. Figures 16-17 of Appendix E show histograms of neural activity over the course of training, demonstrating that activities in the `tanh` network enter the saturation regime during training. This nonlinearity-based compression furnishes another explanation for the observation that training slows down as `tanh` networks enter their compression phase (Shwartz-Ziv & Tishby, 2017): some fraction of inputs have saturated the nonlinearities, reducing backpropagated error gradients.

## 3 INFORMATION PLANE DYNAMICS IN DEEP LINEAR NETWORKS

The preceding section investigates the role of nonlinearity in the observed compression behavior, tracing the source to double-saturating nonlinearities and the binning methodology used to calculate mutual information. However, other mechanisms could lead to compression as well. Even without nonlinearity, neurons could converge to highly correlated activations, or project out irrelevant directions of the input. These phenomena are not possible to observe in our simple three neuron minimal model, as they require multiple inputs and hidden layer activities. To search for these mechanisms, we turn to a tractable model system: deep linear neural networks (Baldi & Hornik (1989); Fukumizu (1998); Saxe et al. (2014)). In particular, we exploit recent results on the generalization dynamics in simple linear networks trained in a student-teacher setup (Seung et al., 1992; Advani & Saxe, 2017). In a student-teacher setting, one "student" neural network learns to approximate the output of another "teacher" neural network. This setting is a way of generating a dataset with interesting structure that nevertheless allows exact calculation of the generalization performance of the network, exact calculation of the mutual information of the representation (without any binning procedure), and, though we do not do so here, direct comparison to the IB bound which is already known for linear Gaussian problems (Chechik et al., 2005).

We consider a scenario where a linear teacher neural network generates input and output examples which are then fed to a deep linear student network to learn (Fig. 3A). Following the formulation of (Advani & Saxe, 2017), we assume multivariate Gaussian inputs $X \sim \mathcal{N}(0, \frac{1}{N_i} I_{N_i})$ and a scalar output $Y$. The output is generated by the teacher network according to $Y = W_0 X + \epsilon_o$, where $\epsilon_o \sim \mathcal{N}(0, \sigma_o^2)$ represents aspects of the target function which cannot be represented by a neural network (that is, the approximation error or bias in statistical learning theory), and the teacher weights $W_o$ are drawn independently from $\mathcal{N}(0, \sigma_w^2)$. Here, the weights of the teacher define the rule to be learned. The signal to noise ratio $\text{SNR} = \sigma_w^2 / \sigma_o^2$ determines the strength of the rule linking inputs to

outputs relative to the inevitable approximation error. We emphasize that the "noise" added to the teacher's output is fundamentally different from the noise added for the purpose of calculating mutual information: $\epsilon_o$ models the approximation error for the task–even the best possible neural network may still make errors because the target function is not representable exactly as a neural network–and is part of the construction of the dataset, not part of the analysis of the student network.

To train the student network, a dataset of $P$ examples is generated using the teacher. The student network is then trained to minimize the mean squared error between its output and the target output using standard (batch or stochastic) gradient descent on this dataset. Here the student is a deep linear neural network consisting of potentially many layers, but where the the activation function of each neuron is simply $f(u) = u$. That is, a depth $D$ deep linear network computes the output $\hat{Y} = W_{D+1} W_D \cdots W_2 W_1 X$. While linear activation functions stop the network from computing complex nonlinear functions of the input, deep linear networks nevertheless show complicated nonlinear learning trajectories (Saxe et al., 2014), the optimization problem remains nonconvex (Baldi & Hornik, 1989), and the generalization dynamics can exhibit substantial overtraining (Fukumizu, 1998; Advani & Saxe, 2017).

Importantly, because of the simplified setting considered here, the true generalization error is easily shown to be

$$E_g(t) = ||W_o - W_{tot}(t)||^2_F + \sigma_o^2 \tag{5}$$

where $W_{tot}(t)$ is the overall linear map implemented by the network at training epoch $t$ (that is, $W_{tot} = W_{D+1} W_D \cdots W_2 W_1$).

Furthermore, the mutual information with the input and output may be calculated exactly, because the distribution of the activity of any hidden layer is Gaussian. Let $T$ be the activity of a specific hidden layer, and let $\bar{W}$ be the linear map from the input to this activity (that is, for layer $l$, $\bar{W} = W_l \cdots W_2 W_1$). Since $T = \bar{W} X$, the mutual information of $X$ and $T$ calculated using differential entropy is infinite. For the purpose of calculating the mutual information, therefore, we assume that Gaussian noise is added to the hidden layer activity, $T = \bar{W} X + \epsilon_{MI}$, with mean 0 and variance $\sigma_{MI}^2 = 1.0$. This allows the analysis to apply to networks of any size, including overcomplete layers, but as before we emphasize that we do not add this noise either during training or testing. With these assumptions, $T$ and $X$ are jointly Gaussian and we have

$$I(T; X) = \log|\bar{W} \bar{W}^T + \sigma_{MI}^2 I_{N_h}| - \log|\sigma_{MI}^2 I_{N_h}| \tag{6}$$

where $|\cdot|$ denotes the determinant of a matrix. Finally the mutual information with the output $Y$, also jointly Gaussian, can be calculated similarly (see Eqns. (22)-(25) of Appendix G).

Fig. 3 shows example training and test dynamics over the course of learning in panel C, and the information plane dynamics in panel D. Here the network has an input layer of 100 units, 1 hidden layer of 100 units each and one output unit. The network was trained with batch gradient descent on a dataset of 100 examples drawn from the teacher with signal to noise ratio of 1.0. The linear network behaves qualitatively like the ReLU network, and does not exhibit compression. Nevertheless, it learns a map that generalizes well on this task and shows minimal overtraining. Hence, in the setting we study here, generalization performance can be acceptable without any compression phase.

The results in (Advani & Saxe (2017)) show that, for the case of linear networks, overtraining is worst when the number of inputs matches the number of training samples, and is reduced by making the number of samples smaller or larger. Fig. 4 shows learning dynamics with the number of samples matched to the size of the network. Here overfitting is substantial, and again no compression is seen in the information plane. Comparing to the result in Fig. 3D, both networks exhibit similar information dynamics with respect to the input (no compression), but yield different generalization performance.

Hence, in this linear analysis of a generic setting, there do not appear to be additional mechanisms that cause compression over the course of learning; and generalization behavior can be widely different for networks with the same dynamics of information compression regarding the input. We note that, in the setting considered here, all input dimensions have the same variance, and the weights of the teacher are drawn independently. Because of this, there are no special directions in the input, and each subspace of the input contains as much information as any other. It is possible that, in real world tasks, higher variance inputs are also the most likely to be relevant to the task (here, have large weights in the teacher). We have not investigated this possibility here.

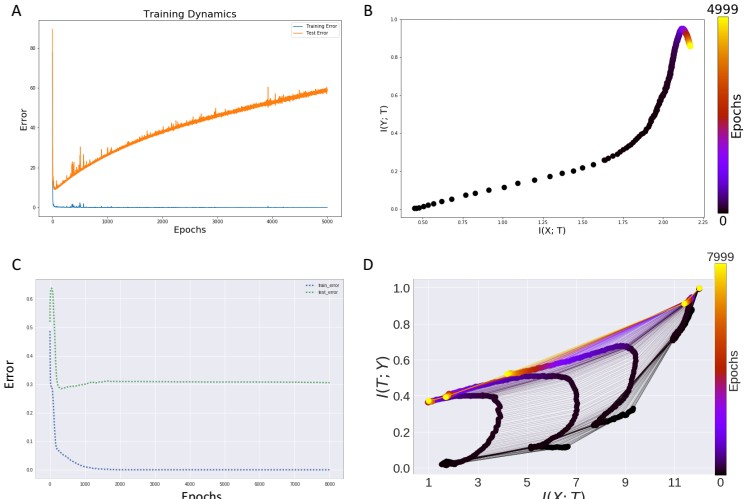

Figure 4: Overtraining and information plane dynamics. (A) Average training and test mean square error for a deep linear network trained with SGD. Overtraining is substantial. Other parameters: $N_i = 100$, P = 100, Number of hidden units = 100, Batch size = 5 (B) Information plane dynamics. No compression is observed, and information about the labels is lost during overtraining. (C) Average train and test accuracy (% correct) for nonlinear $\tanh$ networks exhibiting modest overfitting ($N = 8$). (D) Information plane dynamics. Overfitting occurs despite continued compression.

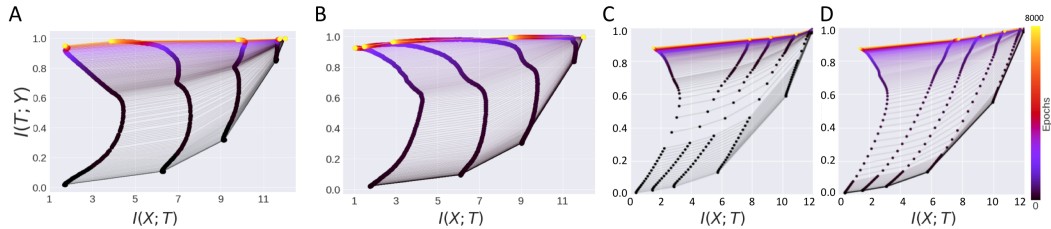

Figure 5: Stochastic training and the information plane. (A) $\tanh$ network trained with SGD. (B) $\tanh$ network trained with BGD. (C) ReLU network trained with SGD. (D) ReLU network trained with BGD. Both random and non-random training procedures show similar information plane dynamics.

To see whether similar behavior arises in nonlinear networks, we trained $\tanh$ networks in the same setting as Section 2, but with 30% of the data, which we found to lead to modest overtraining. Fig. 4C-D shows the resulting train, test, and information plane dynamics. Here the $\tanh$ networks show substantial compression, despite exhibiting overtraining. This establishes a dissociation between behavior in the information plane and generalization dynamics: networks that compress may (Fig. 1A) or may not (Fig. 4C-D) generalize well, and networks that do not compress may (Figs.1B, 3A-B) or may not (Fig. 4A-B) generalize well.

## 4 COMPRESSION IN BATCH GRADIENT DESCENT AND SGD

Next, we test a core theoretical claim of the information bottleneck theory of deep learning, namely that randomness in stochastic gradient descent is responsible for the compression phase. In particular, because the choice of input samples in SGD is random, the weights evolve in a stochastic way during training.

Shwartz-Ziv & Tishby (2017) distinguish two phases of SGD optimization: in the first "drift" phase, the mean of the gradients over training samples is large relative to the standard deviation of the gradients; in the second "diffusion" phase, the mean becomes smaller than the standard deviation of the gradients. The authors propose that compression should commence following the transition

from a high to a low gradient signal-to-noise ratio (SNR), i.e., the onset of the diffusion phase. The proposed mechanism behind this diffusion-driven compression is as follows. The authors state that during the diffusion phase, the stochastic evolution of the weights can be described as a Fokker-Planck equation under the constraint of small training error. Then, the stationary distribution over weights for this process will have maximum entropy, again subject to the training error constraint. Finally, the authors claim that weights drawn from this stationary distribution will maximize the entropy of inputs given hidden layer activity, $H(X|T)$, subject to a training error constraint, and that this training error constraint is equivalent to a constraint on the mutual information $I(T;Y)$ for small training error. Since the entropy of the input, $H(X)$, is fixed, the result of the diffusion dynamics will be to minimize $I(X;T) := H(X) - H(X|T)$ for a given value of $I(T;Y)$ reached at the end of the drift phase.

However, this explanation does not hold up to either theoretical or empirical investigation. Let us assume that the diffusion phase does drive the distribution of weights to a maximum entropy distribution subject to a training error constraint. Note that this distribution reflects stochasticity of weights across different training runs. There is no general reason that a given set of weights sampled from this distribution (i.e., the weight parameters found in one particular training run) will maximize $H(X|T)$, the entropy of inputs given hidden layer activity. In particular, $H(X|T)$ reflects (conditional) uncertainty about inputs drawn from the data-generating distribution, rather than uncertainty about any kind of distribution across different training runs.

We also show empirically that the stochasticity of the SGD is not necessary for compression. To do so, we consider two distinct training procedures: offline stochastic gradient descent (SGD), which learns from a fixed-size dataset, and updates weights by repeatedly sampling a single example from the dataset and calculating the gradient of the error with respect to that single sample (the typical procedure used in practice); and batch gradient descent (BGD), which learns from a fixed-size dataset, and updates weights using the gradient of the total error across all examples. Batch gradient descent uses the full training dataset and, crucially, therefore has no randomness or diffusion-like behavior in its updates.

We trained `tanh` and ReLU networks with SGD and BGD and compare their information plane dynamics in Fig. 5 (see Appendix H for a linear network). We find largely consistent information dynamics in both instances, with robust compression in `tanh` networks for both methods. Thus randomness in the training process does not appear to contribute substantially to compression of information about the input. This finding is consistent with the view presented in Section 2 that compression arises predominantly from the double saturating nonlinearity.

Finally, we look at the gradient signal-to-noise ratio (SNR) to analyze the relationship between compression and the transition from high to low gradient SNR. Fig. 20 of Appendix I shows the gradient SNR over training, which in all cases shows a phase transition during learning. Hence the gradient SNR transition is a general phenomenon, but is not causally related to compression. Appendix I offers an extended discussion and shows gradient SNR transitions without compression on the MNIST dataset and for linear networks.

## 5 Simultaneous Fitting and Compression

Our finding that generalization can occur without compression may seem difficult to reconcile with the intuition that certain tasks involve suppressing irrelevant directions of the input. In the extreme, if certain inputs contribute nothing but noise, then good generalization requires ignoring them. To study this, we consider a variant on the linear student-teacher setup of Section 3: we partition the input $X$ into a set of task-relevant inputs $X_{rel}$ and a set of task-irrelevant inputs $X_{irrel}$, and alter the teacher network so that the teacher's weights to the task-irrelevant inputs are all zero. Hence the inputs $X_{irrel}$ contribute only noise, while the $X_{rel}$ contain signal. We then calculate the information plane dynamics for the whole layer, and for the task-relevant and task-irrelevant inputs separately. Fig. 6 shows information plane dynamics for a deep linear neural network trained using SGD (5 samples/batch) on a task with 30 task-relevant inputs and 70 task-irrelevant inputs. While the overall dynamics show no compression phase, the information specifically about the task-irrelevant subspace does compress over the course of training. This compression process occurs at the same time as the fitting to the task-relevant information. Thus, when a task requires ignoring some inputs, the

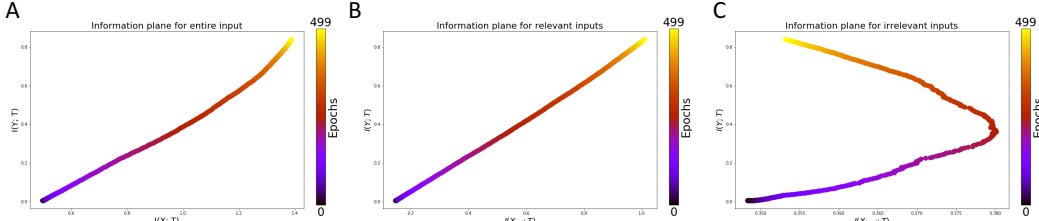

Figure 6: Simultaneous fitting and compression. (A) For a task with a large task-irrelevant subspace in the input, a linear network shows no overall compression of information about the input. (B) The information with the task-relevant subspace increases robustly over training. (C) However, the information specifically about the task-irrelevant subspace does compress after initially growing as the network is trained.

information with these inputs specifically will indeed be reduced; but overall mutual information with the input in general may still increase.

# 6  DISCUSSION

Our results suggest that compression dynamics in the information plane are not a general feature of deep networks, but are critically influenced by the nonlinearities employed by the network. Double-saturating nonlinearities lead to compression, if mutual information is estimated by binning activations or by adding homoscedastic noise, while single-sided saturating nonlinearities like ReLUs do not compress in general. Consistent with this view, we find that stochasticity in the training process does not contribute to compression in the cases we investigate. Furthermore, we have found instances where generalization performance does not clearly track information plane behavior, questioning the causal link between compression and generalization. Hence information compression may parallel the situation with sharp minima: although empirical evidence has shown a correlation with generalization error in certain settings and architectures, further theoretical analysis has shown that sharp minima can in fact generalize well (Dinh et al., 2017). We emphasize that compression still may occur within a subset of the input dimensions if the task demands it. This compression, however, is interleaved rather than in a secondary phase and may not be visible by information metrics that track the overall information between a hidden layer and the input. Finally, we note that our results address the specific claims of one scheme to link the information bottleneck principle with current practice in deep networks. The information bottleneck principle itself is more general and may yet offer important insights into deep networks (Achille & Soatto, 2017). Moreover, the information bottleneck principle could yield fundamentally new training algorithms for networks that are inherently stochastic and where compression is explicitly encouraged with appropriate regularization terms (Chalk et al., 2016; Alemi et al., 2017; Kolchinsky et al., 2017).

## ACKNOWLEDGMENTS

We thank Ariel Herbert-Voss for useful discussions. This work was supported by grant numbers IIS 1409097 and CHE 1648973 from the US National Science Foundation, and by IARPA contract #D16PC00002. Andrew Saxe and Madhu Advani thank the Swartz Program in Theoretical Theoretical Neuroscience at Harvard University. Artemy Kolchinsky and Brendan Tracey would like to thank the Santa Fe Institute for helping to support this research. Artemy Kolchinsky was supported by Grant No. FQXi-RFP-1622 from the FQXi foundation and Grant No. CHE-1648973 from the US National Science Foundation. Brendan Tracey was supported by AFOSR MURI on Multi-Information Sources of Multi-Physics Systems under Award Number FA9550-15-1-0038.

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

## A  LEARNING CURVES FOR tanh AND ReLU NETWORKS

Supplementary Figure 7 shows the learning curves for tanh and ReLU networks depicted in Fig. 1.

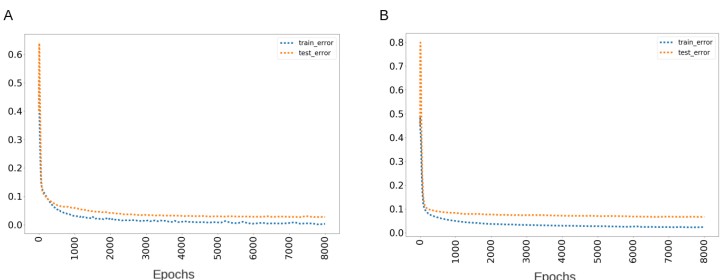

Figure 7: Learning curves for (A) tanh neural network in 1 A and (B) ReLU neural network in 1 B. Both networks show good generalization with regards to the test data.

## B  ROBUSTNESS OF FINDINGS TO MI ESTIMATION METHOD AND NEURAL ACTIVATION FUNCTIONS

This Appendix investigates the generality of the finding that compression is not observed in neural network layers with certain activation functions. Figure 1 of the main text shows example results using a binning-based MI estimator and a nonparametric KDE estimator, for both the tanh and ReLU activation functions. Here we describe the KDE MI estimator in detail, and present extended results on other datasets. We also show results for other activation functions. Finally, we provide entropy estimates based on another nonparametric estimator, the popular k-nearest neighbor approach of Kraskov et al. (2004). Our findings consistently show that double-saturating nonlinearities can yield compression, while single-sided nonlinearities do not.

## B.1    KERNEL DENSITY ESTIMATION OF MI

The KDE approach of Kolchinsky & Tracey (2017); Kolchinsky et al. (2017) estimates the mutual information between the input and the hidden layer activity by assuming that the hidden activity is distributed as a mixture of Gaussians. This assumption is well-suited to the present setting under the following interpretation: we take the input activity to be distributed as delta functions at each example in the dataset, corresponding to a uniform distribution over these specific samples. In other words, we assume that the empirical distribution of input samples is the true distribution. Next, the hidden layer activity $h$ is a deterministic function of the input. As mentioned in the main text and discussed in more detail in Appendix C, without the assumption of noise, this would have infinite mutual information with the input. We therefore assume for the purposes of analysis that Gaussian noise of variance $\sigma^2$ is added, that is, $T = h + \epsilon$ where $\epsilon \sim \mathcal{N}(0, \sigma^2 I)$. Under these assumptions, the distribution of $T$ is genuinely a mixture of Gaussians, with a Gaussian centered on the hidden activity corresponding to each input sample. We emphasize again that the noise $\epsilon$ is added solely for the purposes of analysis, and is not present during training or testing the network. In this setting, an upper bound for the mutual information with the input is (Kolchinsky & Tracey, 2017; Kolchinsky et al., 2017)

$$I(T; X) \leq -\frac{1}{P} \sum_i \log \frac{1}{P} \sum_j \exp\left(-\frac{1}{2} \frac{\|h_i - h_j\|_2^2}{\sigma^2}\right) \tag{7}$$

where $P$ is the number of training samples and $h_i$ denotes the hidden activity vector in response to input sample $i$. Similarly, the mutual information with respect to the output can be calculated as

$$
\begin{aligned}
I(T; Y) \quad &= \quad H(T) - H(T|Y) \tag{8}\\
&\leq \quad -\frac{1}{P} \sum_i \log \frac{1}{P} \sum_j \exp\left(-\frac{1}{2} \frac{\|h_i - h_j\|_2^2}{\sigma^2}\right) \tag{9}\\
&\quad -\sum_l^L p_l \left[-\frac{1}{P_l} \sum_{i, Y_i = l} \log \frac{1}{P_l} \sum_{j, Y_j = l} \exp\left(-\frac{1}{2} \frac{\|h_i - h_j\|_2^2}{\sigma^2}\right)\right] \tag{10}
\end{aligned}
$$

where $L$ is the number of output labels, $P_l$ denotes the number of data samples with output label $l$, $p_l = P_l/P$ denotes the probability of output label $l$, and the sums over $i, Y_i = l$ indicate a sum over all examples with output label $l$.

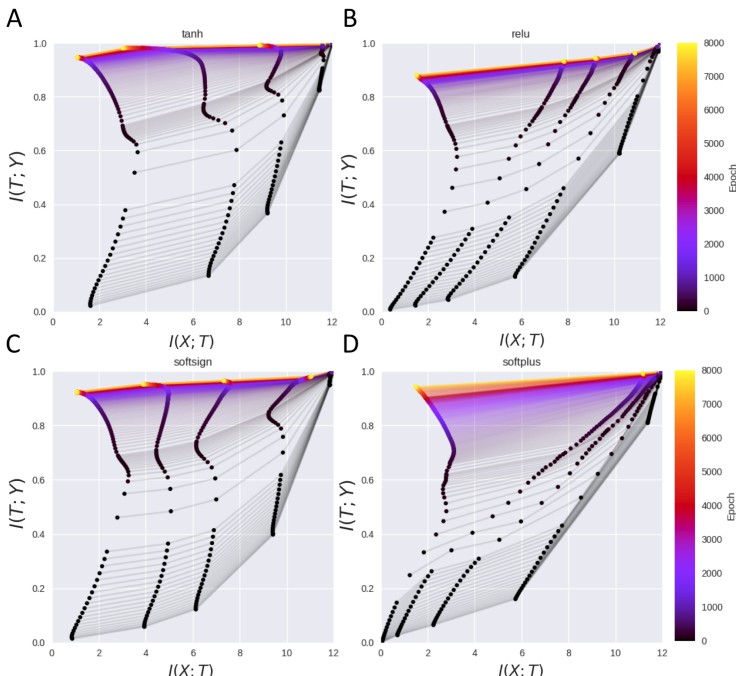

Figure 8: Information plane dynamics for the network architecture and training dataset of Shwartz-Ziv & Tishby (2017), estimated with the nonparametric KDE method of Kolchinsky & Tracey (2017); Kolchinsky et al. (2017) and averaged over 50 repetitions. (A) tanh neural network layers show compression. (B) ReLU neural network layers show no compression. (C) The soft-sign activation function, a double-saturating nonlinearity that saturates more gently than tanh, shows modest compression. (D) The soft-plus activation function, a smoothed version of the ReLU, exhibits no compression. Hence double-saturating nonlinearities exhibit the compression effect while single-saturating nonlinearities do not.

Figure 8A-B shows the result of applying this MI estimation method on the dataset and network architecture of Shwartz-Ziv & Tishby (2017), with MI estimated on the full dataset and averaged over 50 repetitions. Mutual information was estimated using data samples from the test set, and we took the noise variance $\sigma^2 = 0.1$. These results look similar to the estimate derived from binning, with compression in tanh networks but no compression in ReLU networks. Relative to the binning estimate, it appears that compression is less pronounced in the KDE method.

Figure 1C-D of the main text shows the results of this estimation technique applied to a neural network of size $784 - 1024 - 20 - 20 - 20 - 10$ on the MNIST handwritten digit classification dataset. The network was trained using SGD with minibatches of size 128. As before, mutual information was estimated using data samples from the test set, and we took the noise variance $\sigma^2 = 0.1$. The smaller layer sizes in the top three hidden layers were selected to ensure the quality of the kernel density estimator given the amount of data in the test set, since the estimates are more accurate for smaller-dimensional data. Because of computational expense, the MNIST results are from a single training run.

More detailed results for the MNIST dataset are provided in Figure 9 for the tanh activation function, and in Figure 10 for the ReLU activation function. In these figures, the first row shows the evolution of the cross entropy loss (on both training and testing data sets) during training. The second row shows the mutual information between input and the activity of different hidden layers, using the nonparametric KDE estimator described above. The blue region in the second row shows the range of possible MI values, ranging from the upper bound described above (Eq. 10) to the following lower

bound (Kolchinsky & Tracey, 2017),

$$I(T;Y) \geq -\frac{1}{P} \sum_i \log \frac{1}{P} \sum_j \exp \left( -\frac{1}{2} \frac{\|h_i - h_j\|_2^2}{4\sigma^2} \right) \tag{11}$$

$$-\sum_l^L p_l \left[ -\frac{1}{P_l} \sum_{i,Y_i=l} \log \frac{1}{P_l} \sum_{j,Y_j=l} \exp \left( -\frac{1}{2} \frac{\|h_i - h_j\|_2^2}{4\sigma^2} \right) \right]. \tag{12}$$

The third row shows the mutual information between input and activity of different hidden layers, estimated using the binning method (here, the activity of each neuron was discretized into bins of size 0.5). For both the second and third rows, we also plot the entropy of the inputs, $H(X)$, as a dashed line. $H(X)$ is an upper bound on the mutual information $I(X;T)$, and is computed using the assumption of a uniform distribution over the 10,000 testing points in the MNIST dataset, giving $H(X) = \log_2 10000$.

Finally, the fourth row visualizes the dynamics of the SGD updates during training. For each layer and epoch, the green line shows the $\ell_2$ norm of the weights. We also compute the vector of mean updates across SGD minibatches (this vector has one dimension for each weight parameter), as well as the vector of the standard deviation of the updates across SGD minibatches. The $\ell_2$ norm of the mean update vector is shown in blue, and the $\ell_2$ norm of the standard deviation vector is shown in orange. The gradient SNR, computed as the ratio of the norm of the mean vector to the norm of the standard deviation vector, is shown in red. For both the tanh and ReLU networks, the gradient SNR shows a phase transition during training, and the norm of the weights in each layer increases. Importantly, this phase transition occurs despite a lack of compression in the ReLU network, indicating that noise in SGD updates does not yield compression in this setting.

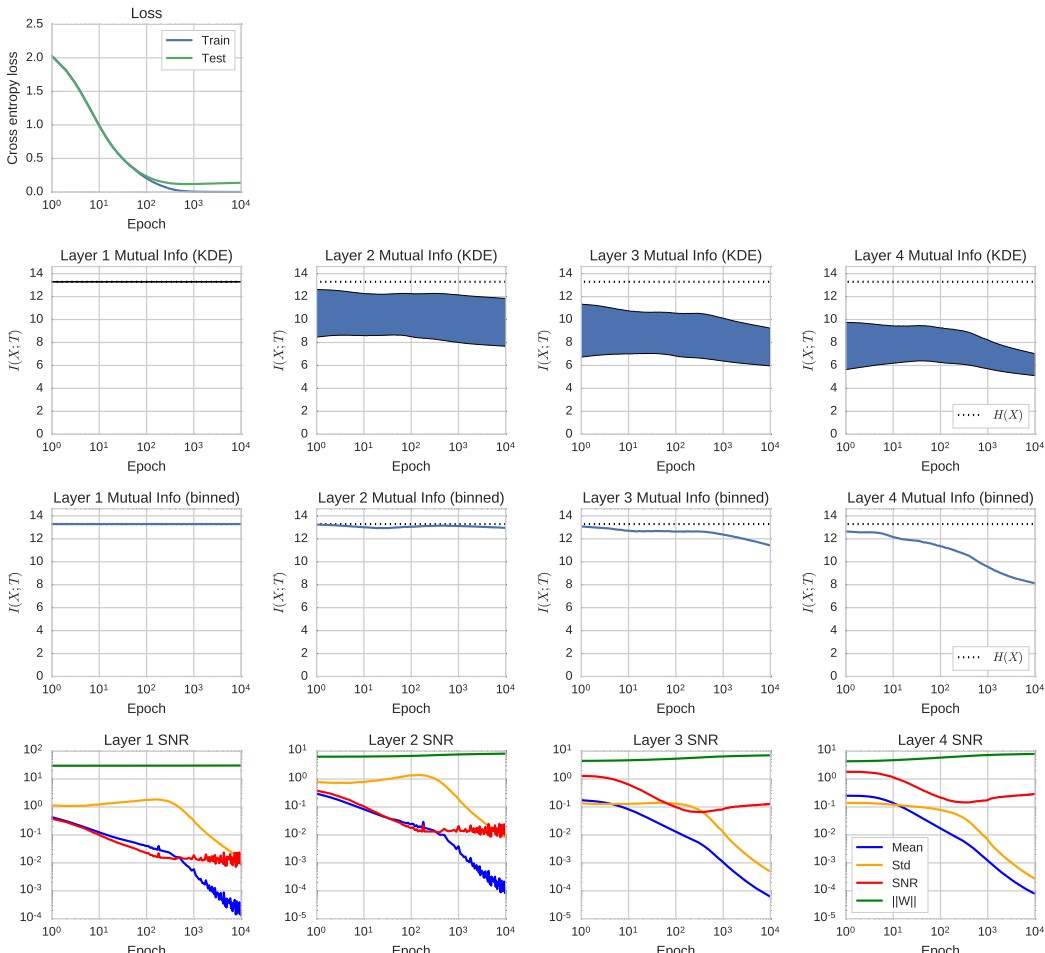

Figure 9: Detailed tanh activation function results on MNIST. Row 1: Loss over training. Row 2: Upper and lower bounds for the mutual information $I(X;T)$ between the input $(X)$ and each layer's activity $(T)$, using the nonparametric KDE estimator (Kolchinsky & Tracey, 2017; Kolchinsky et al., 2017). Dotted line indicates $H(X) = \log_2 10000$, the entropy of a uniform distribution over 10,000 testing samples. Row 3: Binning-based estimate of the mutual information $I(X;T)$, with each neuron's activity discretized using a bin size of 0.5. Row 4: Gradient SNR and weight norm dynamics. The gradient SNR shows a phase transition during training, and the norm of the weights in each layer increases.

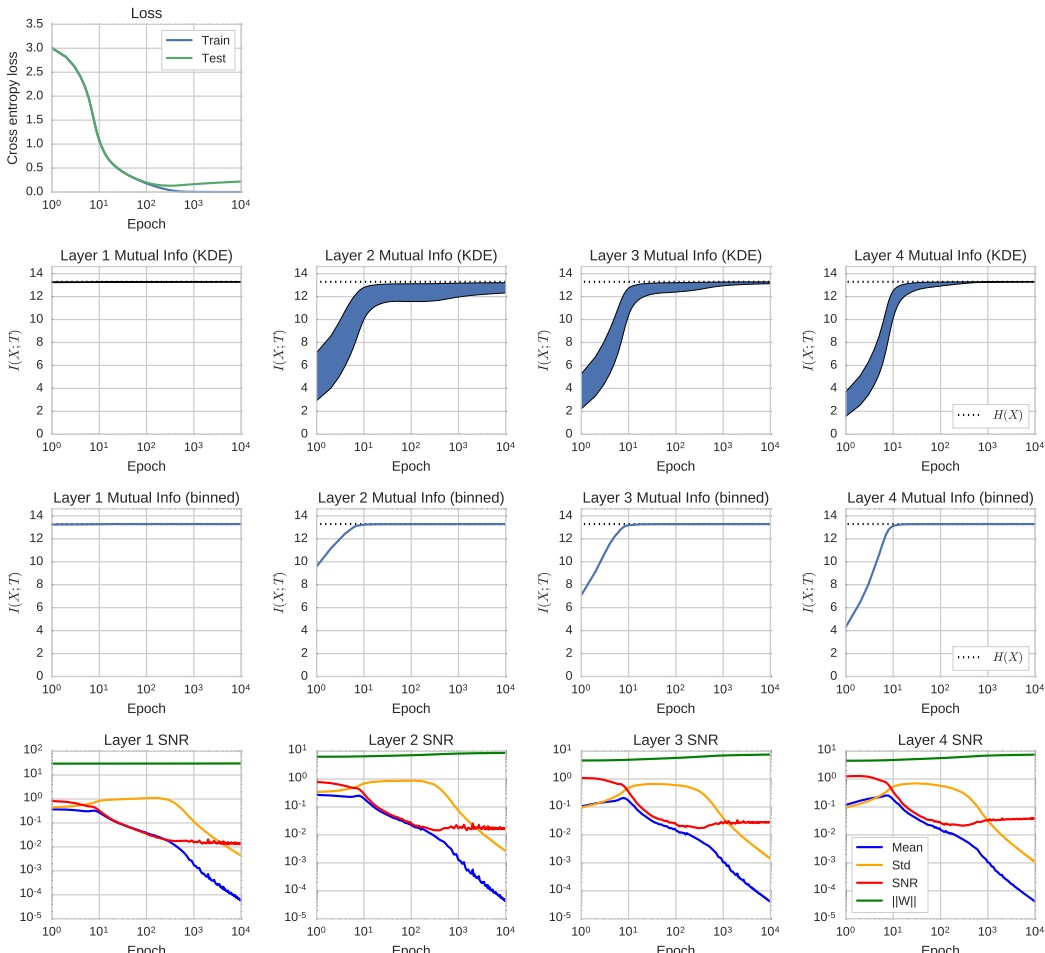

Figure 10: Detailed ReLU activation function results on MNIST. Row 1: Loss over training. Row 2: Upper and lower bounds for the mutual information $I(X;T)$ between the input $(X)$ and each layer's activity $(T)$, using the nonparametric KDE estimator (Kolchinsky & Tracey, 2017; Kolchinsky et al., 2017). Dotted line indicates $H(X) = \log_2 10000$, the entropy of a uniform distribution over 10,000 testing samples. Row 3: Binning-based estimate of the mutual information $I(X;T)$, with each neuron's activity discretized using a bin size of 0.5. Row 4: Gradient SNR and weight norm dynamics. The gradient SNR shows a phase transition during training, and the norm of the weights in each layer increases. Importantly, this phase transition occurs despite a lack of compression in the ReLU network, indicating that noise in SGD updates does not yield compression in this setting.

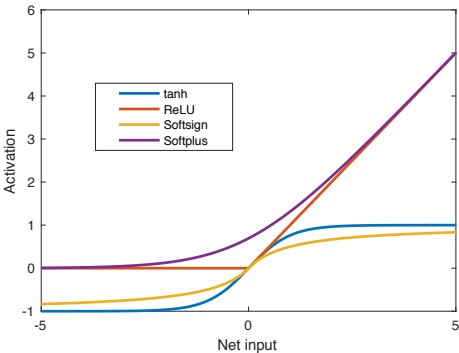

Figure 11: Alternative activation functions.

## B.2 OTHER ACTIVATION FUNCTIONS

Next, in Fig. 8C-D, we show results from the kernel MI estimator from two additional nonlinear activation functions, the softsign function

$$f(x) = \frac{x}{1 + |x|},$$

and the softplus function

$$f(x) = \ln(1 + e^x).$$

These functions are plotted next to $\tanh$ and ReLU in Fig. 11. The softsign function is similar to $\tanh$ but saturates more slowly, and yields less compression than $\tanh$. The softplus function is a smoothed version of the ReLU, and yields similar dynamics with no compression. Because softplus never saturates fully to zero, it retains more information with respect to the input than ReLUs in general.

## B.3 KRASKOV ESTIMATOR

We additionally investigated the widely-used nonparametric MI estimator of Kraskov et al. (2004). This estimator uses nearest neighbor distances between samples to compute an estimate of the entropy of a continuous random variable. Here we focused for simplicity only on the compression phenomenon in the mutual information between the input and hidden layer activity, leaving aside the information with respect to the output (as this is not relevant to the compression phenomenon). Again, without additional noise assumptions, the MI between the hidden representation and the input would be infinite because the mapping is deterministic. Rather than make specific noise assumptions, we instead use the Kraskov method to estimate the entropy of the hidden representations $T$. Note that the entropy of $T$ is the mutual information up to an unknown constant so long as the noise assumption is homoscedastic, that is, $T = h + Z$ where the random variable $Z$ is independent of $X$. To see this, note that

$$
\begin{align}
I(T; X) &= H(T) - H(T|X) \tag{13} \\
&= H(T) - H(Z) \tag{14} \\
&= H(T) - c \tag{15}
\end{align}
$$

where the constant $c = H(Z)$. Hence observing compression in the layer entropy $H(T)$ is enough to establish that compression occurs in the mutual information.

The Kraskov estimate is given by

$$\frac{d}{P} \sum_{i=1}^{P} \log(r_i + \epsilon) + \frac{d}{2} \log(\pi) - \log \Gamma(d/2 + 1) + \psi(P) - \psi(k) \tag{16}$$

where $d$ is the dimension of the hidden representation, $P$ is the number of samples, $r_i$ is the distance to the $k$-th nearest neighbor of sample $i$, $\epsilon$ is a small constant for numerical stability, $\Gamma(\cdot)$ is the

Gamma function, and $\psi(\cdot)$ is the digamma function. Here the parameter $\epsilon$ prevents infinite terms when the nearest neighbor distance $r_i = 0$ for some sample. We took $\epsilon = 10^{-16}$.

Figure 12 shows the entropy over training for $\texttt{tanh}$ and ReLU networks trained on the dataset of and with the network architecture in Shwartz-Ziv & Tishby (2017), averaged over 50 repeats. In these experiments, we used $k = 2$. Compression would correspond to decreasing entropy over the course of training, while a lack of compression would correspond to increasing entropy. Several $\texttt{tanh}$ layers exhibit compression, while the ReLU layers do not. Hence qualitatively, the Kraskov estimator returns similar results to the binning and KDE strategies.

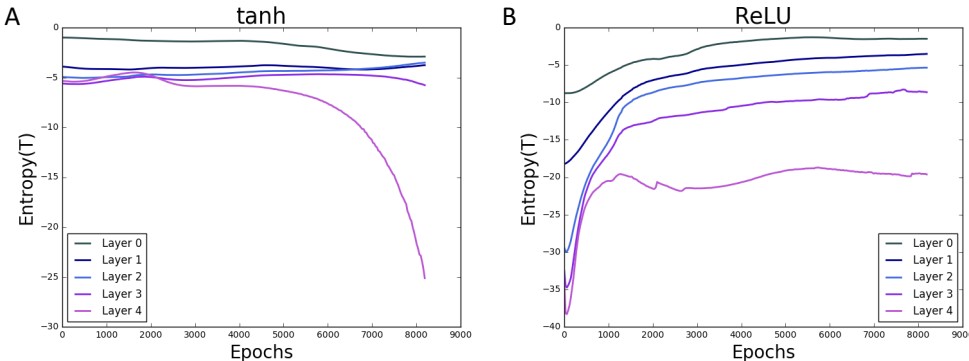

Figure 12: Entropy dynamics over training for the network architecture and training dataset of Shwartz-Ziv & Tishby (2017), estimated with the nonparametric k-nearest-neighbor-based method of Kraskov et al. (2004). Here the x-axis is epochs of training time, and the y-axis plots the entropy of the hidden representation, as calculated using nearest-neighbor distances. Note that in this setting, if $T$ is considered to be the hidden activity plus independent noise, the entropy is equal to the mutual information up to a constant (see derivation in text). Layers 0-4 correspond to the hidden layers of size 10-7-5-4-3. (A) $\texttt{tanh}$ neural network layers can show compression over the course of training. (B) ReLU neural network layers show no compression.

## C    Noise assumptions and discrete vs continuous entropy

A recurring theme in the results reported in this paper is the necessity of noise assumptions to yield a nontrivial information theoretic analysis. Here we give an extended discussion of this phenomenon, and of issues relating to discrete entropy as opposed to continuous (differential) entropy.

The activity of a neural network is often a continuous deterministic function of its input. That is, in response to an input $X$, a specific hidden layer might produce activity $h = f(X)$ for some function $f$. The mutual information between $h$ and $X$ is given by

$$I(h;X) \quad = \quad H(h) - H(h|X). \tag{17}$$

If $h$ were a discrete variable, then the entropy would be given by

$$H(h) = -\sum_{i=1}^{N} p_i \log p_i \tag{18}$$

where $p_i$ is the probability of the discrete symbol $i$, as mentioned in the main text. Then $H(h|X) = 0$ because the mapping is deterministic and we have $I(h;X) = H(h)$.

However $h$ is typically continuous. The continuous entropy, defined for a continuous random variable $Z$ with density $p_Z$ by analogy to Eqn. (18) as

$$H(Z) = -\int p_Z(z) \log p_Z(z) dz, \tag{19}$$

can be negative and possibly infinite. In particular, note that if $p_Z$ is a delta function, then $H(Z) = -\infty$. The mutual information between hidden layer activity $h$ and the input $X$ for continuous $h, X$ is

$$I(h;X) = H(h) - H(h|X). \tag{20}$$

Now $H(h|X) = -\infty$ since given the input $X$, the hidden activity $h$ is distributed as a delta function at $f(X)$. The mutual information is thus generally infinite, so long as the hidden layer activity has finite entropy ($H(h)$ is finite).

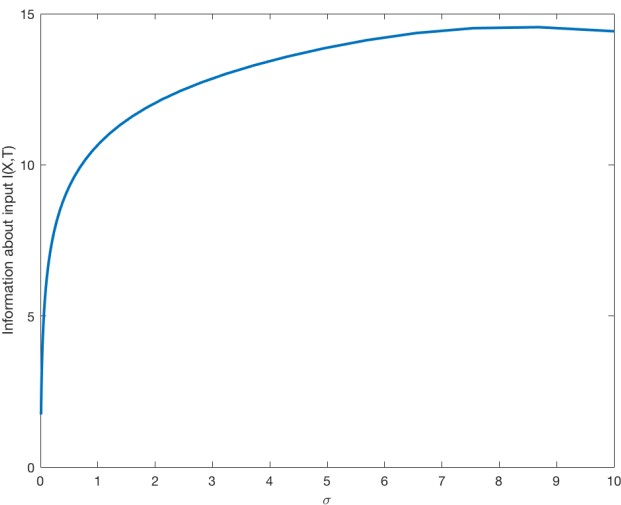

Figure 13: Effect of binning strategy on minimal three neuron model. Mutual information for the simple three neuron model shown in Fig. 2 with bin edges $b_i \in \tanh(\text{linspace}(-50, 50, N))$. In contrast to linear binning, the mutual information continues to increase as weights grow.

To yield a finite mutual information, some noise in the mapping is required such that $H(h|X)$ remains finite. A common choice (and one adopted here for the linear network, the nonparametric kernel density estimator, and the k-nearest neighbor estimator) is to analyze a new variable with additive noise, $T = h + Z$, where $Z$ is a random variable independent of $X$. Then $H(T|X) = H(Z)$ which allows the overall information $I(T; X) = H(T) - H(Z)$ to remain finite. This noise assumption is not present in the actual neural networks either during training or testing, and is made solely for the purpose of calculating the mutual information.

Another strategy is to partition the continuous variable $h$ into a discrete variable $T$, for instance by binning the values (the approach taken in Shwartz-Ziv & Tishby (2017)). This allows use of the discrete entropy, which remains finite. Again, however, in practice the network does not operate on the binned variables $T$ but on the continuous variables $h$, and the binning is solely for the purpose of calculating the mutual information. Moreover, there are many possible binning strategies, which yield different discrete random variables, and different mutual information with respect to the input. The choice of binning strategy is an assumption analogous to choosing a type of noise to add to the representation in the continuous case: because there is in fact no binning in the operation of the network, there is no clear choice for binning methodology. The strategy we use in binning-based experiments reported here is the following: for bounded activations like the $\tanh$ activation, we use evenly spaced bins between the minimum and maximum limits of the function. For unbounded activations like ReLU, we first train the network completely; next identify the minimum and maximum hidden activation over all units and all training epochs; and finally bin into equally spaced bins between these minimum and maximum values. We note that this procedure places no restriction on the magnitude that the unbounded activation function can take during training, and yields the same MI estimate as using infinite equally spaced bins (because bins for activities larger than the maximum are never seen during training).

As an example of another binning strategy that can yield markedly different results, we consider evenly spaced bins in a neuron's net input, rather than its activity. That is, instead of evenly spaced bins in the neural activity, we determine the bin edges by mapping a set of evenly spaced values through the neural nonlinearity. For $\tanh$, for instance, this spaces bins more tightly in the saturation region as compared to the linear region. Figure 13 shows the results of applying this binning strategy

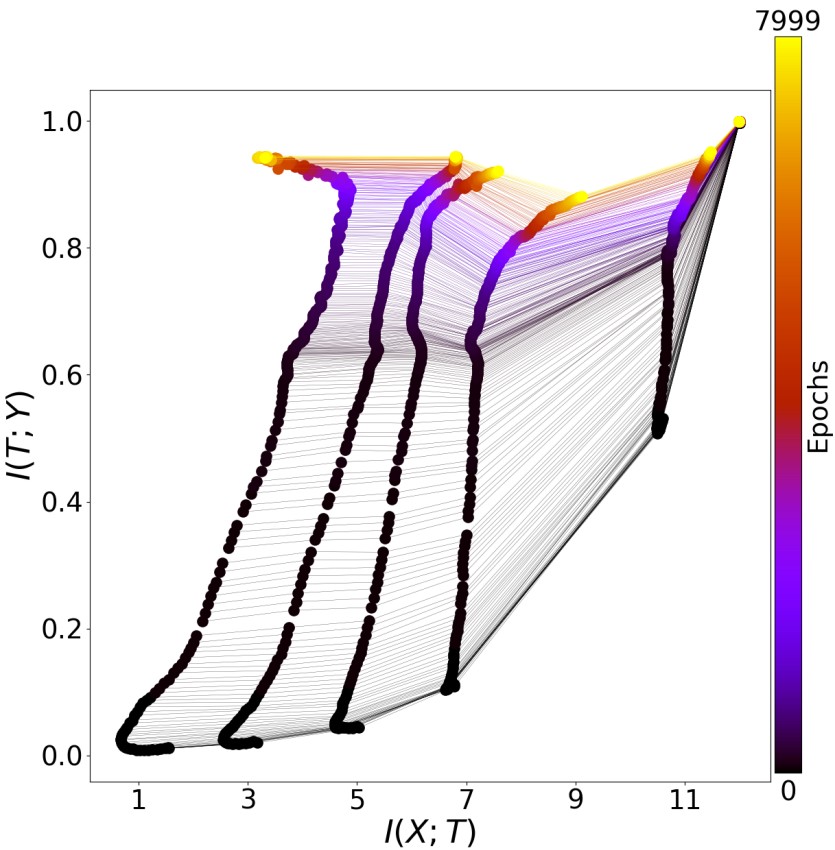

Figure 14: Effect of binning strategy on information plane dynamics. Results for the same $\tanh$ network and training regime as 1A, but with bin edges $b_i \in \tanh(\text{linspace}(-50, 50, N))$. Measured with this binning structure, there is no compression in most layers.

to the minimal three neuron model with $\tanh$ activations. This binning scheme captures more information as the weights of the network grow larger. Figure 14 shows information plane dynamics for this binning structure. The $\tanh$ network no longer exhibits compression. (We note that the broken DPI in this example is an artifact of performing binning only for analysis, as discussed below).

Any implementation of a neural network on digital hardware is ultimately of finite precision, and hence is a binned, discrete representation. However, it is a very high resolution binning compared to that used here or by Shwartz-Ziv & Tishby (2017): single precision would correspond to using roughly $2^{32}$ bins to discretize each hidden unit's activity, as compared to the 30-100 used here. If the binning is fine-grained enough that each input $X$ yields a different binned activity pattern $h$, then $H(h) = \log(P)$ where $P$ is the number of examples in the dataset, and there will be little to no change in information during training. As an example, we show in Fig. 15 the result of binning at full machine precision.

Finally, we note two consequences of the assumption of noise/binning for the purposes of analysis. First, this means that the data processing inequality (DPI) does not apply to the noisy/binned mutual information estimates. The DPI states that information can only be destroyed through successive transformations, that is, if $X \rightarrow h_1 \rightarrow h_2$ form a Markov chain, then $I(X; h_1) \geq I(X; h_2)$ (see, eg, Tishby & Zaslavsky (2015)). Because noise is added only for the purpose of analysis, however, this does not apply here. In particular, for the DPI to apply, the noise added at lower layers would have to propagate through the network to higher layers. That is, if the transformation from hidden layer 1 to hidden layer 2 is $h_2 = f(h_1)$ and $T_1 = h_1 + Z_1$ is the hidden layer activity after adding noise, then the DPI would hold for the variable $\tilde{T}_2 = f(T_1) + Z_2 = f(h_1 + Z_1) + Z_2$, not the quantity

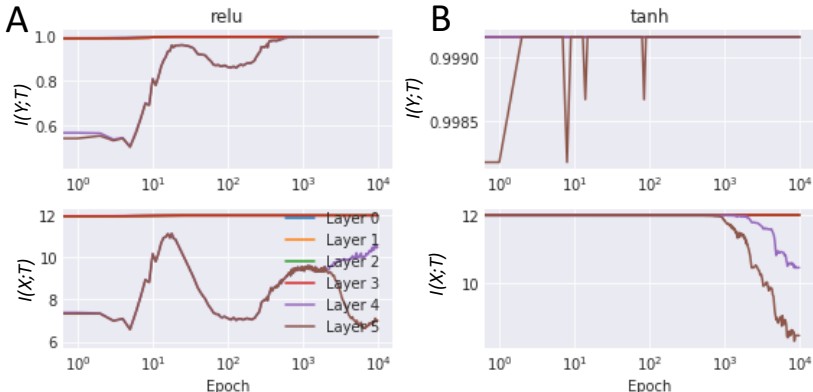

Figure 15: Effect of binning at full machine precision. (A) ReLU network. (B) $\tanh$ network. Information in most layers stays pinned to $\log_2(P) = 12$. Compression is only observed in the highest and smallest layers near the very end of training, when the saturation of $\tanh$ is strong enough to saturate machine precision.

$T_2 = h_2 + Z_2 = f(h_1) + Z_2$ used in the analysis. Said another way, the Markov chain for $T_2$ is $X \to h_1 \to h_2 \to T_2$, not $X \to h_1 \to T_1 \to T_2$, so the DPI states only that $I(X; h_1) \geq I(X; T_2)$.

A second consequence of the noise assumption is the fact that the mutual information is no longer invariant to invertible transformations of the hidden activity $h$. A potentially attractive feature of a theory based on mutual information is that it can allow for comparisons between different architectures: mutual information is invariant to any invertible transformation of the variables, so two hidden representations could be very different in detail but yield identical mutual information with respect to the input. However, once noise is added to a hidden representation, this is no longer the case: the variable $T = h + Z$ is not invariant to reparametrizations of $h$. As a simple example, consider a minimal linear network with scalar weights $w_1$ and $w_2$ that computes the output $\hat{y} = w_2 w_1 X$. The hidden activity is $h = w_1 X$. Now consider the family of networks in which we scale down $w_1$ and scale up $w_2$ by a factor $c \neq 0$, that is, these networks have weights $\tilde{w}_1 = w_1/c$ and $\tilde{w}_2 = cw_2$, yielding the exact same input-output map $\hat{y} = \tilde{w}_2 \tilde{w}_1 X = cw_2(w_1/c)X = w_2 w_1 X$. Because they compute the same function, they necessarily generalize identically. However after introducing the noise assumption the mutual information is

$$I(T; X) = \log\left(w_1^2/c^2 + \sigma_{MI}^2\right) - \log\left(\sigma_{MI}^2\right) \tag{21}$$

where we have taken the setting in Section 3 in which $X$ is normal Gaussian, and independent Gaussian noise of variance $\sigma_{MI}^2$ is added for the purpose of MI computation. Clearly, the mutual information is now dependent on the scaling $c$ of the internal layer, even though this is an invertible linear transformation of the representation. Moreover, this shows that networks which generalize identically can nevertheless have very different mutual information with respect to the input when it is measured in this way.

## D   WEIGHT NORMS OVER TRAINING

Our argument relating neural saturation to compression in mutual information relies on the notion that in typical training regimes, weights begin small and increase in size over the course of training. We note that this is a virtual necessity for a nonlinearity like $\tanh$, which is linear around the origin: when initialized with small weights, the activity of a $\tanh$ network will be in this linear regime and the network can only compute a linear function of its input. Hence a real world nonlinear task can only be learned by increasing the norm of the weights so as to engage the $\tanh$ nonlinearity on some examples. This point can also be appreciated from norm-based capacity bounds on neural networks, which show that, for instance, the Rademacher complexity of a neural network with small weights must be low (Bartlett & Mendelson, 2002; Neyshabur et al., 2015). Finally, as an empirical matter, the networks trained in this paper do in fact increase the norm of their weights over the course of

training, as shown by the green lines in Figure 20 for $\mathrm{tanh}$ and ReLU networks in the training setting of Shwartz-Ziv & Tishby (2017); Figures 9 and 10 for the MNIST networks; and Figure 21 for a linear network.

# E   HISTOGRAMS OF NEURAL ACTIVATIONS

Supplementary Figures 16 and 17 show histograms of neural activities over the course of training in $\mathrm{tanh}$ and ReLU networks respectively.

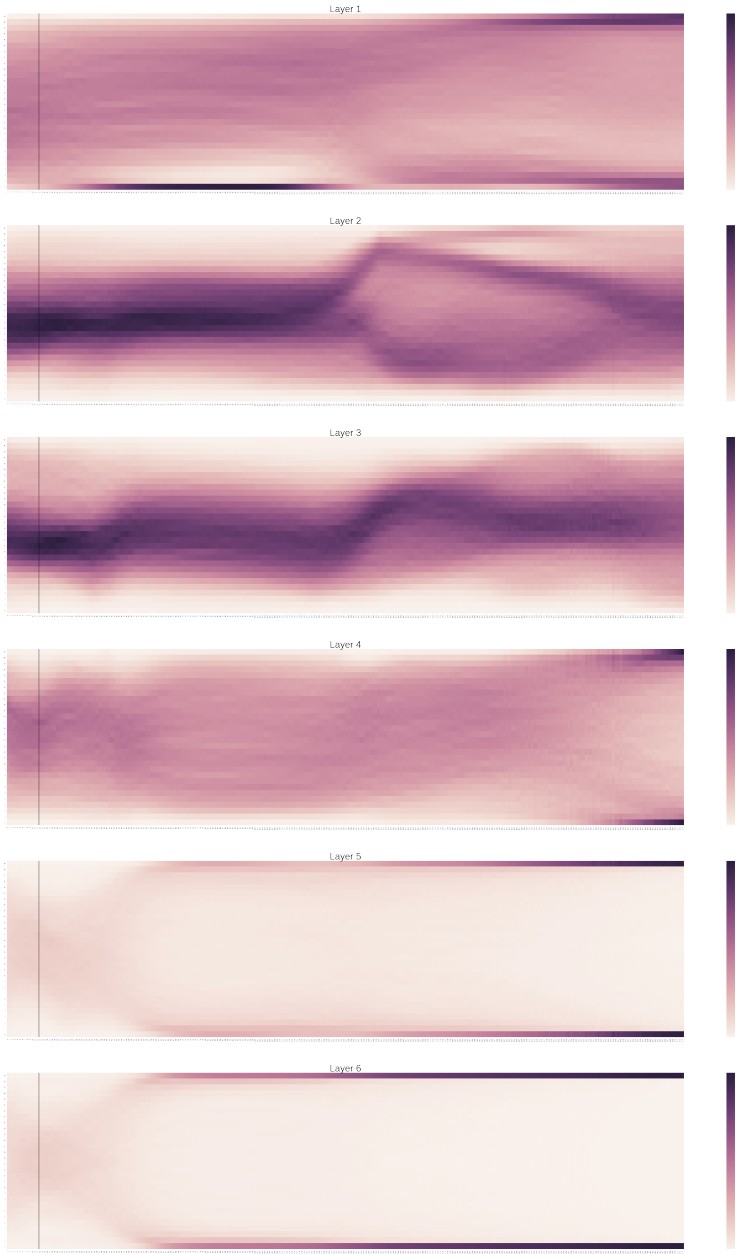

Figure 16: Histogram of neural activities in a $\mathrm{tanh}$ network during training. The final three layers eventually saturate in the top and bottom bins corresponding to the saturation limits of the $\mathrm{tanh}$ activation function, explaining the compression observed in $\mathrm{tanh}$. x-axis: training time in epochs. y-axis: Hidden activity bin values from lowest to highest. Colormap: density of hidden layer activities across all input examples.

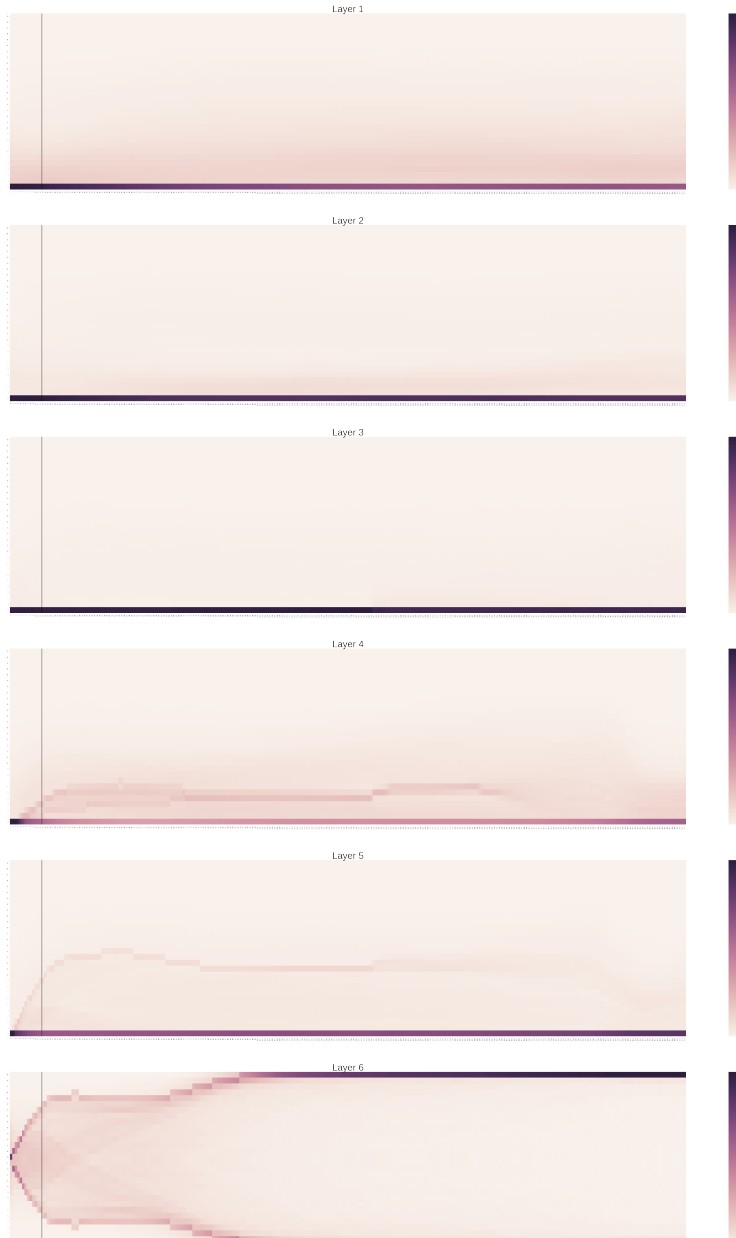

Figure 17: Histogram of neural activities in a ReLU network during training. ReLU layers 1-5 have a roughly constant fraction of activities at zero, corresponding to instances where the ReLU is off; the nonzero activities disperse over the course of training without bound, yielding higher entropy distributions. The sigmoid output layer 6 converges to its saturation limits, and is the only layer that compresses during training (c.f. Fig. 1B). x-axis: training time in epochs. y-axis: Hidden activity value. Colormap: density of hidden layer activities across all input examples.

## F    INFORMATION PLANE DYNAMICS IN DEEPER LINEAR NETWORKS

Supplementary Figure 18 shows information plane dynamics for a deep neural network with five hidden layers each containing 50 hidden units.

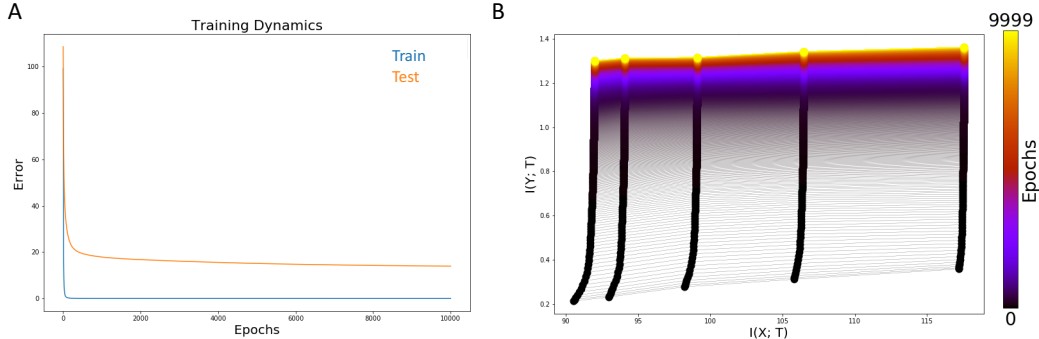

Figure 18: Information plane dynamics in a deep linear neural network. (A) Train and test error during learning. (B) Information plane dynamics. No compression is visible.

## G   LINEAR MUTUAL INFORMATION CALCULATION

For the linear setting considered here, the mutual information between a hidden representation $T$ and the output $Y$ may be calculated using the relations

$$H(Y) = \frac{N_o}{2} \log(2\pi e) + \frac{1}{2} \log|W_o W_o^T + \sigma_o^2 I_{N_o}|, \tag{22}$$

$$H(T) = \frac{N_h}{2} \log(2\pi e) + \frac{1}{2} \log|\bar{W}\bar{W}^T + \sigma_{MI}^2 I_{N_h}|, \tag{23}$$

$$H(Y;T) = \frac{N_o + N_h}{2} \log(2\pi e) + \frac{1}{2} \log \begin{vmatrix} \bar{W}\bar{W}^T + \sigma_{MI}^2 I_{N_h} & \bar{W}W_o^T, \\ W_o \bar{W}^T & W_o W_o^T + \sigma_o^2 I_{N_h} \end{vmatrix}, \tag{24}$$

$$I(Y;T) = H(Y) + H(T) - H(Y;T). \tag{25}$$

## H   STOCHASTIC VS BATCH TRAINING

Figure 19 shows information plane dynamics for stochastic and batch gradient descent learning in a linear network. Randomness in the training process does not dramatically alter the information plane dynamics.

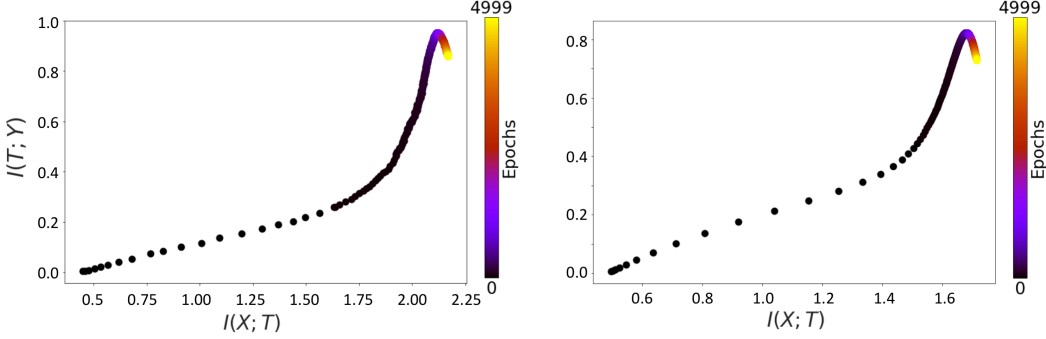

Figure 19: Effect of stochastic training in linear networks. (A) Information plane dynamics for stochastic gradient descent in a linear network (same setting as Fig. 4). (B) Information plane dynamics for batch gradient descent.

## I   GRADIENT SNR PHASE TRANSITION

The proposed mechanism of compression in Shwartz-Ziv & Tishby (2017) is noise arising from stochastic gradient descent training. The results in Section 4 of the main text show that compression

still occurs under batch gradient descent learning, suggesting that in fact noise in the gradient updates is not the cause of compression. Here we investigate a related claim, namely that during training, networks switch between two phases. These phases are defined by the ratio of the mean of the gradient to the standard deviation of the gradient across training examples, called the gradient signal-to-noise ratio. In the first "drift" phase, the SNR is high, while in the second "diffusion" phase the SNR is low. Shwartz-Ziv & Tishby (2017) hypothesize that the drift phase corresponds to movement toward the minimum with no compression, while the diffusion phase corresponds to a constrained diffusion in weight configurations that attain the optimal loss, during which representations compress. However, two phases of gradient descent have been described more generally, sometimes known as the transient and stochastic phases or search and convergence phases (Murata, 1998; Chee & Toulis, 2017), suggesting that these phases might not be related specifically to compression behavior.

In Fig. 20 we plot the gradient SNR over the course of training for the tanh and ReLU networks in the standard setting of Shwartz-Ziv & Tishby (2017). In particular, for each layer $l$ we calculate the mean and standard deviation as

$$m_l = \left\| \left\langle \frac{\partial E}{\partial W_l} \right\rangle \right\|_F \tag{26}$$

$$s_l = \left\| \text{STD}\left( \frac{\partial E}{\partial W_l} \right) \right\|_F \tag{27}$$

where $\langle \cdot \rangle$ denotes the mean and $STD(\cdot)$ denotes the element-wise standard deviation across all training samples, and $\|\cdot\|_F$ denotes the Frobenius norm. The gradient SNR is then the ratio $m_l/s_l$. We additionally plot the norm of the weights $\|W_l\|_F$ over the course of training.

Both tanh and ReLU networks yield a similar qualitative pattern, with SNR undergoing a step-like transition to a lower value during training. Figures 9 and 10, fourth row, show similar plots for MNIST-trained networks. Again, SNR undergoes a transition from high to low over training. Hence the two phase nature of gradient descent appears to hold across the settings that we examine here. Crucially, this finding shows that the SNR transition is not related to the compression phenomenon because ReLU networks, which show the gradient SNR phase transition, do not compress.

Finally, to show the generality of the two-phase gradient SNR behavior and its independence from compression, we develop a minimal model of this phenomenon in a three neuron linear network. We consider the student-teacher setting of Fig. 3 but with $N_i = N_h = 1$, such that the input and hidden layers have just a single neuron (as in the setting of Fig. 2). Here, with just a single hidden neuron, clearly there can be no compression so long as the first layer weight increases over the course of training. Figure 21AC shows that even in this simple setting, the SNR shows the phase transition but the weight norm increases over training. Hence again, the two phases of the gradient are present even though there is no compression. To intuitively understand the source of this behavior, note that the weights are initialized to be small and hence early in learning all must be increased in magnitude, yielding a consistent mean gradient. Once the network reaches the vicinity of the minimum, the mean weight change across all samples by definition goes to zero. The standard deviation remains finite, however, because on some specific examples error could be improved by increasing or decreasing the weights–even though across the whole dataset the mean error has been minimized.

Hence overall, our results show that a two-phase structure in the gradient SNR occurs in all settings we consider, even though compression occurs only in a subset. The gradient SNR behavior is therefore not causally related to compression dynamics, consistent with the view that saturating nonlinearities are the primary source of compression.

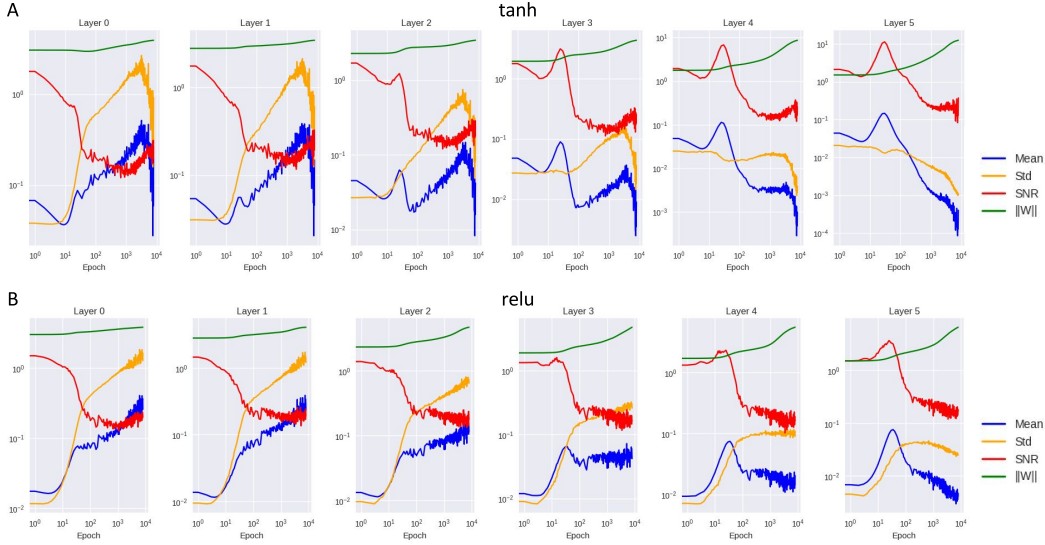

Figure 20: Gradient SNR phase transition. (A) tanh networks trained in the standard setting of Shwartz-Ziv & Tishby (2017) show a phase transition in every layer. (B) ReLU networks also show a phase transition in every layer, despite exhibiting no compression.

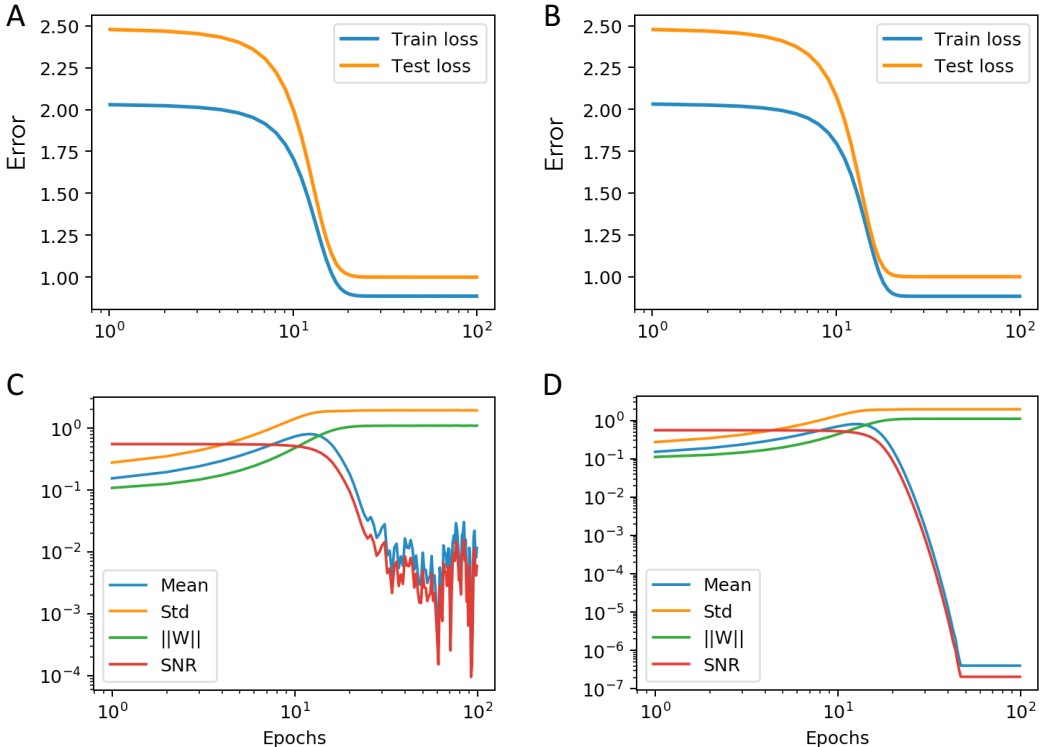

Figure 21: Minimal model exhibiting gradient SNR phase transition. Here a three neuron linear network (architecture $1 - 1 - 1$) learns to approximate a teacher. Other parameters are teacher $SNR = 1$, number of training samples $P = 100$, learning rate .001. Left column: (A) The loss over training with SGD (minibatch size 1). (C) The resulting gradient SNR dynamics. Right column: (B) The loss over training with BGD. (D) The resulting gradient SNR dynamics averaging over all training samples (not minibatches, see text).

