# OpenReview forum: "On the Information Bottleneck Theory of Deep Learning"
_ICLR.cc/2018/Conference — Accept (Poster)_

### Official Review · AnonReviewer3 · 2017-11-27
**An ongoing debate**

**Rating:** 6
**Confidence:** 2

**Review:**

This paper presents a study on the Information Bottleneck (IB) theory of deep learning, providing results in contrasts to the main theory claims. According to the authors, the IB theory suggests that the network generalization is mainly due to a ‘compression phase’ in the information plane occurring after a ‘fitting phase’ and that the ‘compression phase’ is due to the stochastic gradient decent (SDG). Instead, the results provided by this paper show that: the generalization can happen even without compression; that SDG is not the primary factor in compression; and that the compression does not necessarily occur after the ‘fitting phase’. Overall, the paper tackles the IB theory claims with consistent methodology, thus providing substantial arguments against the IB theory.

The main concern is that the paper is built to argue against another theoretical work, raising a substantial discussion with the authors of the IB theory. This paper should carefully address all the raised arguments in the main text.

There are, moreover, some open questions that are not fully clear in this contribution:
1)	To evaluate the mutual information in the ReLu networks (sec. 2) the authors discretize the output activity in their range. Should the non-linearity of ReLu be considered as a form of compression? Do you check the ratio of ReLus that are not active during training or the ratio of inputs that fall into the negative domain of each ReLu?
2)	Since one of today common topics is the training of deep neural networks with lower representational precision, could the quantization error due to the low precision be considered as a form of noise inserted in the network layers that influences the generalization performance in deep neural networks?
3)	What are the main conclusions or impact of the present study in the theory of neural networks? Is it the authors aim to just demonstrate that the IB theory is not correct? Perhaps, the paper should empathize the obtained results not just in contrast to the other theory, but proactively in agreement with a new proposal.

Finally, a small issue comes from the Figures that need some improvement. In most of the cases (Figure 3 C, D; Figure 4 A, B, C; Figure 5 C, D; Figure 6) the axes font is too small to be read. Figure 3C is also very unclear.

---

> ### Author Response · Authors · 2018-01-04
> **Response to Reviewer 3**
>
> Please also see our comments to all reviewers above.
>
> -This paper presents a study on the Information Bottleneck (IB) theory of deep learning, providing results in contrasts to the main theory claims. According to the authors, the IB theory suggests that the network generalization is mainly due to a ‘compression phase’ in the information plane occurring after a ‘fitting phase’ and that the ‘compression phase’ is due to the stochastic gradient decent (SDG). Instead, the results provided by this paper show that: the generalization can happen even without compression; that SDG is not the primary factor in compression; and that the compression does not necessarily occur after the ‘fitting phase’. Overall, the paper tackles the IB theory claims with consistent methodology, thus providing substantial arguments against the IB theory.
>
> Thank you!
>
> -The main concern is that the paper is built to argue against another theoretical work, raising a substantial discussion with the authors of the IB theory. This paper should carefully address all the raised arguments in the main text.
>
> The revision now addresses these arguments in the main text. We believe the conclusions in our original submission still stand, and are now supported by additional experiments.
>
> -There are, moreover, some open questions that are not fully clear in this contribution:
> -1) To evaluate the mutual information in the ReLu networks (sec. 2) the authors discretize the output activity in their range. Should the non-linearity of ReLu be considered as a form of compression? Do you check the ratio of ReLus that are not active during training or the ratio of inputs that fall into the negative domain of each ReLu?
>
> Our discretization does consider the nonlinearity of ReLU, which could in principle lead to compression if ReLUs tended to inactivate over the course of training. However they do not seem to in practice, which can be seen from the histograms of activity over training in Fig. 17. The bottom-most bin contains zero, the ReLU saturation value. There is no consistent trend in the number of saturated ReLU activations over training, with most layers ending up about where they started, with neurons inactive on roughly 50% of examples.
>
> -2) Since one of today common topics is the training of deep neural networks with lower representational precision, could the quantization error due to the low precision be considered as a form of noise inserted in the network layers that influences the generalization performance in deep neural networks?
>
> Thank you for the suggestion, we now point to this possibility in the discussion. For networks which explicitly incorporate noise in their architecture (either through quantization or noise injection), the broader information bottleneck theory may apply and yield potentially new training algorithms. Our point in this paper is that the specific claims of the information bottleneck theory of deep learning, which attempt to explain the performance of “vanilla” deep networks with no quantization or noise, do not in fact explain the generalization performance of these networks.
>
> -3) What are the main conclusions or impact of the present study in the theory of neural networks? Is it the authors aim to just demonstrate that the IB theory is not correct? Perhaps, the paper should empathize the obtained results not just in contrast to the other theory, but proactively in agreement with a new proposal.
>
> There are a variety of theories (several cited in our introduction) which may be consistent with all of the results reported in this paper. Most directly, the results in Advani & Saxe, 2017 successfully account for generalization behavior in the linear models we study. However even there, it remains to be seen how those ideas might apply to deep nonlinear networks. It is outside the scope of this paper to provide strong support for any one of these theories, as singling out one theory as better would require experiments designed to specifically test them, which must be left for future work. Our aim rather was to carefully and fairly inspect an exciting and, it seemed to us, promising theory, and the result turned out to be somewhat negative. In our view, negative results are a critical component of a healthy research ecosystem, and on occasion science advances through falsification. The impact of the present study on the theory of neural networks is to help narrow the field of plausible candidate theories. We expect our results to be important to researchers currently building off of the information bottleneck theory of deep learning.
>
> -Finally, a small issue comes from the Figures that need some improvement. In most of the cases (Figure 3 C, D; Figure 4 A, B, C; Figure 5 C, D; Figure 6) the axes font is too small to be read. Figure 3C is also very unclear.
>
> We apologize for this issue, we have increased the size of several figures and are working towards a revision with the rest corrected.

---

### Official Review · AnonReviewer2 · 2017-11-28
**An interesting and probably controversial paper, discussing the limitations of the information bottleneck theory for deep learning.**

**Rating:** 7
**Confidence:** 3

**Review:**

The authors address the issue of whether the information bottleneck (IB) theory can provide insight into the working of deep networks. They show, using some counter-examples, that the previous understanding of IB theory and its application to deep networks is limited.

PROS: The paper is very well written and makes its points very clearly. To the extent of my knowledge, the content is original. Since it clearly elucidates the limitations of IB theory in its ability to analyse deep networks, I think it is a significant
contribution worthy of acceptance. The experiments are also well designed and executed.

CONS: On the downside, the limitations exposed are done so empirically, but the underlying theoretical causes are not explored (although this could be potentially because this is hard to do). Also, the paper exposes the limitations of another paper published in a non-peer reviewed location (arXiv) which potentially limits its applicability and significance.

Some detailed comments:

In section 2, the influence of binning on how the mutual information is calculated should be made clear. Since the comparison is between a bounded non-linearity and an unbounded one, it is not self-evident how the binning in the latter case should be done. A justification for the choice made for binning the relu case would be helpful.

In the same section, it is claimed that the dependence of the mutual information I(X; T) on the magnitude of the weights of the network explains why a tanh non-linearity shows the compression effect (non-monotonicity vs I(X; T)) in the information plane dynamics. But the claim that large weights are required for doing anything useful is unsubstantiated and would benefit from having citations to papaers that discuss this issue. If networks with small weights are able to learn most datasets, the arguments given in this section wouldn't be applicable in its entirety.

Additionally, figures that show the phase plane dynamics for other non-linearities e.g. relu+ or sigmoid, should be added, at
least in the supplementary section. This is important to complete the overall picture of how the compression effect depends on having specific activation functions.

In section 3, a sentence or two should be added to describe what a "teacher-student setup" is, and how it is relevant/interesting.

Also in section 3, the cases where batch gradient descent is used and where stochastic gradient descent is used should be
pointed out much more clearly. It is mentioned in the first line of page 7 that batch gradient descent is used, but it is not
clear why SGD couldn't have been used to keep things consistent. This applies to figure 4 too.

In section 4, it seems inconsistent that the comparison of SGD vs BGD is done using linear network as opposed to a relu network which is what's used in Section 2. At the least, a comparison using relu should be added to the supplementary section.

Minor comments
The different figure styles using in Fig 4A and C that have the same quantities plotted makes it confusing.
An additional minor comment on the figures: some of the labels are hard to read on the manuscript.

---

> ### Author Response · Authors · 2018-01-04
> **Response to Reviewer 2, part I**
>
> Please also see our comments to all reviewers above.
>
> -PROS: The paper is very well written and makes its points very clearly. To the extent of my knowledge, the content is original. Since it clearly elucidates the limitations of IB theory in its ability to analyse deep networks, I think it is a significant contribution worthy of acceptance. The experiments are also well designed and executed.
>
> Thank you!
>
> -CONS: On the downside, the limitations exposed are done so empirically, but the underlying theoretical causes are not explored (although this could be potentially because this is hard to do). Also, the paper exposes the limitations of another paper published in a non-peer reviewed location (arXiv) which potentially limits its applicability and significance.
>
> While we agree that we have not been able to prove theoretically that, for instance, ReLUs will not compress, we do believe we have elucidated some of the theoretical causes: we present a minimal three neuron model that exhibits the compression phenomenon and give an explicit formula for the binning-based MI estimate; and we give exact calculations of the MI for the linear case, for which the generalization behavior is known. Finally, we now directly discuss the fact that SGD does not necessarily behave like BGD plus additive noise (and hence there is no stochastic relaxation to a Gibbs distribution).
>
> Although the information bottleneck theory of deep learning has appeared only as an arXiv paper, it has achieved attention through video lectures and articles in the popular press. Most importantly from our perspective, researchers are actively attempting to build new methods off of the ideas in the information bottleneck theory, and we believe our results could be significant to those efforts—this, in our view, is the main value in our present work.

---

> > ### Author Response · Authors · 2018-01-04
> > **Response to Reviewer 2, part II**
> >
> > -Some detailed comments:
> >
> > -In section 2, the influence of binning on how the mutual information is calculated should be made clear. Since the comparison is between a bounded non-linearity and an unbounded one, it is not self-evident how the binning in the latter case should be done. A justification for the choice made for binning the relu case would be helpful.
> >
> > For ReLU, we simply space bins up to the largest activation value encountered over the course of training (this method places no a priori assumption on how large the activations might grow, and is equivalent to having bins stretching to infinity since all larger bins would never be used and have probability zero). We have added an extended discussion to the appendix which, in addition to these points, shows the results of alternative binning strategies.
> >
> > -In the same section, it is claimed that the dependence of the mutual information I(X; T) on the magnitude of the weights of the network explains why a tanh non-linearity shows the compression effect (non-monotonicity vs I(X; T)) in the information plane dynamics. But the claim that large weights are required for doing anything useful is unsubstantiated and would benefit from having citations to papaers that discuss this issue. If networks with small weights are able to learn most datasets, the arguments given in this section wouldn't be applicable in its entirety.
> >
> > We have now included an appendix which justifies this claim. First, we note that nonlinearities like tanh are linear near the origin. Hence small weights place activities in this linear regime and the network can only compute a linear function of the input. As essentially all real world tasks are nonlinear, it is a virtual necessity for the weights to increase until the tanh nonlinearities saturate on some examples. More generally, we cite Rademacher complexity bounds which depend on the norm of the weights (implying that small weight networks can represent only simple functions). Finally, as an emiprical matter, we show that for the tanh, ReLU, and linear networks considered in this paper the weight norms increase in every layer over training.
> >
> > -Additionally, figures that show the phase plane dynamics for other non-linearities e.g. relu+ or sigmoid, should be added, at least in the supplementary section. This is important to complete the overall picture of how the compression effect depends on having specific activation functions.
> >
> > Thank you, we have now added two more nonlinearities (softplus and softsign) to the appendix, which also show similar results.
> >
> >  -In section 3, a sentence or two should be added to describe what a "teacher-student setup" is, and how it is relevant/interesting. Also in section 3, the cases where batch gradient descent is used and where stochastic gradient descent is used should be pointed out much more clearly. It is mentioned in the first line of page 7 that batch gradient descent is used, but it is not clear why SGD couldn't have been used to keep things consistent. This applies to figure 4 too.
> >
> > We have now more fully described the student-teacher scenario, and more carefully labeled the batch size in our experiments (though we note that it made little difference on the information plane dynamics in our hands).
> >
> > -In section 4, it seems inconsistent that the comparison of SGD vs BGD is done using linear network as opposed to a relu network which is what's used in Section 2. At the least, a comparison using relu should be added to the supplementary section.
> >
> > We now use the ReLU network in the main text, and have placed the linear network result in the appendix.
> >
> > -Minor comments: The different figure styles using in Fig 4A and C that have the same quantities plotted makes it confusing. An additional minor comment on the figures: some of the labels are hard to read on the manuscript.
> >
> > We apologize for these issues, we intend to submit another revision with larger figure captions and consistent plotting styles.

---

### Official Review · AnonReviewer1 · 2017-11-28
**important contribution to deep learning theory**

**Rating:** 7
**Confidence:** 3

**Review:**

A thorough investigation on Info Bottleneck and deep learning, nice to read with interesting experiments and references. Even though not all of the approach is uncontroversial (as the discussion shows), the paper contributes to much needed theory of deep learning rather than just another architecture.
Estimating the mutual information could have been handled in a more sophisticated way (eg using a Kraskov estimator rather than simple binning), and given that no noise is usually added the discussion about noise and generalisation doesn't seem to make too much sense to me.

It would have been good to see a discussion whether another measurement that would be useful for single-sided saturating nonlinearities that do show a compression (eg information from a combination of layers), from learnt representations that are different to representations learnt using double-sided nonlinearities.

Regarding the finite representation of units (as in the discussion) it might be helpful to also consider an implementation of a network with arbitrary precision arithmetic as an additional experiment.

Overall I think it would be nice to see the paper accepted at the very least to continue the discussion.

---

> ### Author Response · Authors · 2018-01-04
> **Response to Reviewer 1**
>
> Please also note our comments to all reviewers above.
>
> -A thorough investigation on Info Bottleneck and deep learning, nice to read with interesting experiments and references. Even though not all of the approach is uncontroversial (as the discussion shows), the paper contributes to much needed theory of deep learning rather than just another architecture.
>
> Thanks for the encouraging comments!
>
> -Estimating the mutual information could have been handled in a more sophisticated way (eg using a Kraskov estimator rather than simple binning), and given that no noise is usually added the discussion about noise and generalisation doesn't seem to make too much sense to me.
>
> We now include the Kraskov estimator as well as a nonparametric KDE estimator, which show similar results to the binning-based estimate.
>
> We have revised the text to clarify that the `''noise'' in the student-teacher section on generalization is fundamentally different from the noise added to representations for analysis. It represents approximation error (i.e., aspects of the target function which even the best neural network of a given architecture cannot model), and is part of generating an interesting dataset based on a teacher. The noise added to representations for analysis, by contrast, is an assumption which affects the student network itself, and is not part of the operation of the student network in practice.
>
> -It would have been good to see a discussion whether another measurement that would be useful for single-sided saturating nonlinearities that do show a compression (eg information from a combination of layers), from learnt representations that are different to representations learnt using double-sided nonlinearities.
>
> So long as hidden activities are continuous, we believe that MI between the input and multiple layers simultaneously should show similar dynamics. Given our results, it seems that single-sided saturating nonlinearities do not in general compress, and this would carry through to measures that combine multiple layers (because these layers form a Markov chain).
>
> -Regarding the finite representation of units (as in the discussion) it might be helpful to also consider an implementation of a network with arbitrary precision arithmetic as an additional experiment. Overall I think it would be nice to see the paper accepted at the very least to continue the discussion.
>
> Thank you for the suggestion, we considered doing an experiment with arbitrary precision but were able to rule out this concern through another route: if noise in batch gradient descent from numerical precision causes the weights to converge to a Gibbs distribution, and this in turn causes compression, then we should see compression in ReLU or linear networks trained with BGD. However we do not, as we now show in Fig. 5D, which makes this explanation unlikely in our eyes. Moreover, even the noise in SGD appears insufficient to cause compression for ReLU or linear networks, and hence is unlikely to be the source of compression more generally.

---

### Public Comment · (anonymous) · 2017-11-11
**Part I of the response of Naftali Tishby & Ravid Shwartz-Ziv**

1. We would like to thank the authors for taking the effort to repeat and verify many of our numerical experiments. Basically, this paper confirms our theory and strengthen it. Unfortunately, the paper ignores much of our theoretical and experimental results and is flawed and misleading in many ways.

2. In the archive papers and much more in the YouTube talks [https://www.youtube.com/watch?v=bLqJHjXihK8&t=912s , https://www.youtube.com/watch?v=FSfN2K3tnJU&t=5781s] which followed it, we give two independent theoretical arguments on (1) why and how the compression of the representation dramatically improves generalization, and (2) how the stochastic relaxation, due to either noise of the SGD by mini batches, OR a noisy training energy surface which effectively adds smaller similar noise also to BGD, push the weights distribution to a Gibbs measure in the training error.  This is an old argument used in the statistical mechanics of learning 25 years ago, and is used today by many (e.g. Poggio).
We then argue that this weight Gibbs distribution leads directly (essentially through Bayes rule) to the IB optimal encoders of the layers. These theoretical results are the real core of our theory, not the numerical simulations.

3. Also showed in these talks some of our newer simulations, which include much larger and different problems (MNIST, CIFAR-10 with RelU nonlinearties, different architectures, CNN, Linear networks, etc.).
In ALL these networks we observe essentially the same picture: at least the last hidden layer first improves generalization error (which is actually proved in my Berlin talk [20:53] to be DIRECTLY bounded by the mutual information on Y) by fitting the training data and adding more information on the inputs, and then further improve generalization by compressing the representation and “forget” the irrelevant details of the inputs. During both these phases of training the information on the relevant components of the input increases monotonically, as we show in our paper and nicely verified in the last section of this paper. One can of course have input compression without generalization, when the training size is too small to keep the homogeneity of the cover. This we clearly show in the paper and talk ([28:34] top left), as follows from the theory.

4. We also showed in the talk [32:11]and paper that there are clearly and directly two phases of the gradients distribution. First, high SNR gradients follow by a sharp flip to low SNR gradients, which corresponds to the slow saturation of the training error. This clear gradients phase transition, which we see with all types of non-linearities and architectures, beautifully corresponds to the “knee” between memorization and compression phases in the information plane.
This gradient phase transition was reported by several other people. See e.g. https://medium.com/intuitionmachine/the-peculiar-behavior-of-deep-learning-loss-surfaces-330cb741ec17.
This can be explained as done by Poggio in his theory 3 paper, or by Riccardo Zecchina and his coworkers using statistical mechanics.

5. This transition has little to do with the saturation of the nonlinearities, but mainly with the complex nature of the training error surfaces in high dimension.   The saturation of the non-linearities is directly related the “collapsing gradients” phenomenon, which is well understood and led to the usage of RelU and other non-saturating non-linearities.
Our compression phase happens BEFORE this saturation, and the compression is not a consequence of the saturation. Indeed, as we also noted, some of the units are pushed to the hard binary limit eventually, which makes the partition of the encoder harder. This can only enhance the compression, as also shown in this paper (rather inconsistent with other claims in the paper).

See also part 2.

---

> ### Public Comment · ~Naftali_Tishby1 · 2017-11-11
> **Part 2 of the response of Naftali Tishby and Ravid Shwartz-Ziv**
>
> Part 2 of the response of Naftali Tishby and Ravid Shwartz-Ziv
>
> 6.  The main flaw/misconception of this work is in the estimate the mutual information in the RelU case. For RelU Networks often people use some regularization which limits the weight magnitude. Anyway, even without regularization, there is some distribution on the values of the units with a finite controlled variance, and the CDF of this distribution is the effective nonlinearity. This CDF should be binned equally (max entropy binning) as we do with the saturated tanh nonlinearity. This binning, or noise if you prefer, is NOT arbitrary! It has to to do with the ALWAYS FINITE precision of the units. The mutual information is bounded by both the inputs and layer entropies, and is always finite due to this inherent discretization of the units.  When doing this correct quantization on the RelUs,  we obtain , as shown in the talk [34:18], exactly the same compression phase as with saturated units.
>
> In fact binning is only the simplest way to estimate the MI. For RelU units (and for much larger networks) we estimated it using more sophisticated parametric methods such as mixture of Gaussian.
> There is a lot of detailed literature on how to estimate MI in DNNs in practice. See e.g. https://arxiv.org/abs/1705.02436 .﻿
>
> 7. We have much to say about the linear analysis. It should be compared, as said in the paper, to the Linear Gaussian IB (GIB). Then one could nicely see the convergence to the GIB information curve through compression (projections to the CCA space). In general, however, linear networks don’t capture the most interesting aspects of deep learning, in our opinion.﻿

---

> > ### Author Response · Authors · 2017-11-15
> > **Response to Part II**
> >
> > 6. The main flaw/misconception of this work is in the estimate the mutual information in the RelU case. RelU Networks can’t converge without some regularization which limits the weight magnitude. This induces some distribution on the values of the units with a finite controlled variance, and the CDF of this distribution is the effective nonlinearity. This CDF should be binned equally (max entropy binning) as we do with the saturated tanh nonlinearity. This binning, or noise if you prefer, is NOT arbitrary! It has to to do with the ALWAYS FINITE precision of the units. The mutual information is bounded by both the inputs and layer entropies, and is always finite due to this inherent discretization of the units. When doing this correct quantization on the RelUs, we obtain , as shown in the talk [34:18], exactly the same compression phase as with saturated units.
> >
> > In our simulations on ReLU networks in this paper, we have used the binning strategy described in “Opening the Black Box of Deep Neural Networks via information.” At a minimum, our results speak to the empirical methodology used in that paper (we note that max entropy binning, contrary to the comment, is not what is done in the “Opening the Black Box…” paper for tanh). Additionally we point out that in many cases ReLU networks can converge without regularization as has been found in simulations and described in a recent work examining generalization dynamics in neural networks https://arxiv.org/abs/1710.03667.
> >
> > Despite the fact that our binning strategy is the same one used in the paper in question, in response to the authors’ suggestion we have investigated other binning strategies as well (we emphasize that estimating mutual information in high dimensions is a notoriously difficult task and all binning or parametric methods are approximate, hence none are ‘correct’ or ‘incorrect’). Regarding max entropy binning, we note that this aims to use every bin equally frequently; and therefore the information is constant across training, so long as nonlinearities are invertible. Because of this we thought the even bin spacing used in the original paper was a fairer comparison. In particular, max entropy binning of the full joint CDF would typically yield constant information with respect to the input for linear and tanh networks, contradicting the results in the paper (intuitively, more bins are spaced in the saturation regime of the tanh units). It is possible that tanh units could show compression if they saturate hard enough due to machine precision, but this further highlights the importance of saturation and nonlinearity. We note that calculating the bin spacings from the marginal CDFs for each neuron separately would be far from the true max entropy binning of the joint CDF because it assumes independence between neurons. For ReLU networks, max entropy binning may not be able to achieve equal frequency bins because the nonlinearity is not invertible (any negative preactivations map exactly to zero). However this clearly shows that compression, if it is measured under this method, would be due to the impact of saturation and nonlinearity; and furthermore, shows the sensitivity of observed compression in the information plane to the method of MI estimation. More broadly, estimating mutual information is a difficult task in high dimensions regardless of the estimation technique employed. Thankfully, we note that our linear networks permit exact calculation of the MI, sidestepping the necessity to estimate MI entirely. For the linear case we find that there is no compression, providing at least one counterexample showing that a two phase fitting/compression dynamic is not a universal phenomenon. We believe our results provide reason for caution that every deep network will undergo a two phase fitting/compression dynamic.

---

> > > ### Author Response · Authors · 2017-11-15
> > > **Response to Part II (continued)**
> > >
> > > 7. We have much to say about the linear analysis. It should be compared, as said in the paper, to the Linear Gaussian IB (GIB). Then one could nicely see the convergence to the GIB information curve through compression (projections to the CCA space). In general, however, linear networks don’t capture the most interesting aspects of deep learning, in our opinion.﻿
> > >
> > >
> > > Linear neural networks are studied here because they are a simple system (albeit high-dimensional and with non-linear dynamics) which we can understand fully and where compression cannot occur due to saturation of nonlinear units, a very important issue as we point out in this work. Linear networks also sidestep the issue of MI estimation methods, because the MI can be calculated exactly. The fact that we do not reliably observe compression as defined in “Opening the Black Box of Deep Neural Networks via information” in linear systems where many of the input dimensions are irrelevant appears to be an important issue which would need to be addressed by such a theory before being applied to complex systems where there could be multiple reasons for compression. We do observe compression of irrelevant dimensions as should be expected, but interestingly do not see a compression phase when we consider all dimensions (as is done in “Opening the Black Box ...”), which suggests that saturation of nonlinearities seems to be crucial to observe the compression phase in the way it has been defined. Finally, our linear results reiterate that these phenomena arise even with batch training where there is no noise in the training procedure, and hence a stochastic relaxation is not responsible for the resulting information plane dynamics.
> > >
> > > Overall, with respect to the theory, we have shown that compression does not happen due to stochastic relaxation. And with respect to the empirical claims in “Opening the Black Box…”, we have shown that the observed compression results arise primarily from the double saturating nonlinearities and method of MI estimation, not stochasticity in SGD. We believe these are important results to communicate to the community.

---

> ### Author Response · Authors · 2017-11-15
> **Response to Part I**
>
> Thank you for the comments, we have carefully investigated them and responded in full below.
>
> 2. In the archive papers and much more in the YouTube talks [https://www.youtube.com/watch?v=bLqJHjXihK8&t=912s , https://www.youtube.com/watch?v=FSfN2K3tnJU&t=5781s] which followed it, we give two independent theoretical arguments on (1) why and how the compression of the representation dramatically improves generalization, and (2) how the stochastic relaxation, due to either noise of the SGD by mini batches, OR a noisy training energy surface which effectively adds smaller similar noise also to BGD, push the weights distribution to a Gibbs measure in the training error. This is an old argument used in the statistical mechanics of learning 25 years ago, and is used today by many (e.g. Poggio). We then argue that this weight Gibbs distribution leads directly (essentially through Bayes rule) to the IB optimal encoders of the layers. These theoretical results are the real core of our theory, not the numerical simulations.
>
> We disagree with a core theoretical result of this theory, namely that stochastic relaxation is responsible for the compression phase. There are two proposed ways that a stochastic relaxation could arise: first, SGD could behave eventually like a constrained diffusion; and second, a “noisy training energy surface” could effectively add noise to BGD. With respect to SGD, we have shown that the compression phase occurs even without it by using BGD without adding noise. Moreover, the theory relies on the noise in SGD acting like a constrained diffusion, whereas the behavior of SGD is in fact far from this because updates are highly correlated. This was also pointed out in a recent ICLR submission [“On the inductive bias of stochastic gradient descent”  https://openreview.net/forum?id=HyWrIgW0W ] which states: “SGD does not even converge in the classical sense: we show that the most likely trajectories of SGD for deep networks do not behave like Brownian motion around critical points. Instead, they resemble closed loops with deterministic components.”
>
> With respect to the suggestion that there is a “noisy training energy surface which effectively adds noise also to BDG,” in our batch gradient descent setting the training energy surface contains no noise that would cause the weights to converge to a Gibbs distribution. The statistical mechanics of learning papers (eg, Seung, Sompolinsky, & Tishby (1992)) explicitly add isotropic noise to the gradient to obtain Langevin dynamics and a Gibbs distribution over weights as is done in Eqn. 9 of Poggio’s Theory 3 paper, which we point out does not claim an equivalence between SGD and these Langevin dynamics. We emphasize that adding noise to the learning rules is not the standard practice in deep networks. For our batch GD setting there is no noise in the training dynamics, no Gibbs distribution on the weights, and yet nevertheless we observe nearly identical dynamics in the information plane.
>
> Using the simple three neuron model, we show clearly that nonlinearity and the binning procedure can cause compression in this instance. This is our main point, which addresses a core claim of the information bottleneck theory of deep learning: compression does not appear to happen through a stochastic relaxation because (a) the randomness in SGD does not behave like a diffusion, (b) we observe identical compression even with true batch GD, where there is no noise and no stochastic relaxation, and (c) we have identified a simple mechanism that explains the observed empirical results based on the neural nonlinearity. We disagree with the statements “the diffusion phase mostly adds random noise to the weights, and they evolve like Wiener processes...” and “The stochasticity of SGD methods is usually motivated as a way of escaping local minima of the training error. In this paper we give it a new, perhaps much more important role: it generates highly efficient internal representations through compression by diffusion” for the reasons outlined above.

---

> > ### Author Response · Authors · 2017-11-15
> > **Response to Part I (continued)**
> >
> > 3. Also showed in these talks some of our newer simulations, which include much larger and different problems (MNIST, CIFAR-10 with RelU nonlinearties, different architectures, CNN, Linear networks, etc.). In ALL these networks we observe essentially the same picture: at least the last hidden layer first improves generalization error (which is actually proved in my Berlin talk [20:53] to be DIRECTLY bounded by the mutual information on Y) by fitting the training data and adding more information on the inputs, and then further improve generalization by compressing the representation and “forget” the irrelevant details of the inputs. During both these phases of training the information on the relevant components of the input increases monotonically, as we show in our paper and nicely verified in the last section of this paper. One can of course have input compression without generalization, when the training size is too small to keep the homogeneity of the cover. This we clearly show in the paper and talk ([28:34] top left), as follows from the theory.
> >
> > Our results differ with these findings for ReLU and linear networks. Regarding compression in  “at least the last hidden layer,” if this is the final softmax output layer, the results would be consistent with ours. If this is the final ReLU/linear layer just before the softmax output, does this mean that compression is not observed in lower layers? Again, in our simulations, we see no compression in any ReLU or linear layer, only at the final softmax output (which, as we show, can be explained by the double saturating nonlinearity in the final layer). Because they are analytically tractable, our results in linear networks in particular show that no compression in any layer occurs in these instances. Regarding the comment “information on the relevant components of the input increases monotonically, as we show in our paper”, we are not sure what this refers to in “Opening the Black Box…”. We note that we can define relevant input components in our case only because we specify the data generation process. However, this is not possible for datasets like the ones used in “Opening the Black Box…”.

---

> > > ### Author Response · Authors · 2017-11-15
> > > **Response to Part I (continued #2)**
> > >
> > > 4. We also showed in the talk [32:11]and paper that there are clearly and directly two phases of the gradients distribution. First, high SNR gradients follow by a sharp flip to low SNR gradients, which corresponds to the slow saturation of the training error. This clear gradients phase transition, which we see with all types of non-linearities and architectures, beautifully corresponds to the “knee” between memorization and compression phases in the information plane. This gradient phase transition was reported by several other people. See e.g. https://medium.com/intuitionmachine/the-peculiar-behavior-of-deep-learning-loss-surfaces-330cb741ec17. This can be explained as done by Poggio in his theory 3 paper, or by Riccardo Zecchina and his coworkers using statistical mechanics.
> > >
> > > Please see our response to comment 5, which also addresses these comments.
> > >
> > > 5. This transition has little to do with the saturation of the nonlinearities, but mainly with the complex nature of the training error surfaces in high dimension. The saturation of the non-linearities is directly related the “collapsing gradients” phenomenon, which is well understood and led to the usage of RelU and other non-saturating non-linearities. Our compression phase happens BEFORE this saturation, and the compression is not a consequence of the saturation. Indeed, as we also noted, some of the units are pushed to the hard binary limit eventually, which makes the partition of the encoder harder. This can only enhance the compression, as also shown in this paper (rather inconsistent with other claims in the paper).
> > >
> > > These two phases of stochastic gradient descent are general, known variously as the transient and stochastic phases or search and convergence phases, and are not a result of the complex nature of the training error surface in high dimensions (see, eg, Murata, 1998; Chee & Toulis, 2017). For instance, our simple 3 neuron model is not high dimensional but nevertheless shows this behavior, which has a straightforward origin: the transient phase corresponds to forgetting the initialization (if weights are initialized to be small, all must be increased, yielding a consistent mean gradient); the stochastic phase corresponds to oscillating in the vicinity of the minimum (when weights are large and the training error is near zero, different examples need the weights to be increased or decreased, yielding higher variance gradients). However, these two phases are not the cause of the observed compression. These phases happen for ReLU and linear networks (plots we will add to the appendix), where no compression is observed. And we emphasize that we have shown directly that compression is indeed a consequence of saturation and the approach to saturation for the tanh networks (note that compression due to the tanh nonlinearity can happen well before the ‘hard binary limit’). That is, we agree that SGD has two phases in general; but we disagree that these phases are causally connected to the compression observed, which we have shown to be due to the nonlinearity and binning methodology.

---

> ### Public Comment · ~Aaron_Schumacher1 · 2017-12-07
> **Question re: "This gradient phase transition was reported by several other people."**
>
> The linked Medium post mentions "the gradient phase transition" because it is reporting Shwartz-Ziv & Tishby's paper. I don't see an independent verification there. The post does reference some other papers; does one of them contain such a verification?

---

### Public Comment · ~Naftali_Tishby1 · 2017-11-28
**Data falsifying the claims of this ICLR submission all together.**

Naftali Tishby and Ravid Shwartz Ziv

Final public comment on the ICLR 2018 Conference Paper852
On the Information Bottleneck Theory of Deep Learning
This “paper” attacks our work through the following flawed and misleading statements:

1.	That the compression of the representation (reducing I(T:X)) is due to the saturated non-linearity and is not appear with other non-linearity (RelU’s in particular).

The authors don’t know how to estimate mutual information correctly. When properly done, there essentially the same fitting and compression phases with RelU’s and any other network we examined:

Here are the Information Plane trajectories for the CIFAR-10 CNN networks with RelU’s non-linearity as shown in our presentations:

Figure 1 (see attachment)
One can easily see the two phases and the phase transition between them (where I(X;T) has its maximum).


2.	“that there is no evident causal connection between compression and generalization”

We rigorously proved that compression leads to dramatic improvement in generalization, providing that the partitions remained homogenous to the label probability. In fact we argue that any bit of representation compression (under these conditions) is effective as doubling the size of the training data! Here is the sketch of our proof as given in our presentations:

Figure 2 (see attachment)
3.	“that the compression is unrelated to the noisy (low SNR) phase of the gradients”, as we claim.

Below are some figures that clearly show the precise relation between the beginning of the compression phase (argmax I(X;T) for the last hidden layer (green line on left) and the gradient-SNR transition (blue line on right):

Figure 3 (see attachment)

Moreover, when changing the min-batch size (from 32 to 4000) both transitions move together in perfect linear relationship (left). In fact we show (right) that the full batch case (BGD of the “paper”) lies on the same line (green point) which suggests that the reported compression here is exactly the same phenomena, for much weaker gradient noise (as we claimed).

We believe these facts nullify the arguments given in this “paper” all together.

See attachment with figures:
https://www.dropbox.com/s/6aotykw6py37z1h/Naftali%20Tishby%20and%20Ravid%20Shwartz%20Ziv-final%20comment.pdf?dl=0

---

> ### Author Response · Authors · 2018-01-04
> **Response to Tishby & Shwartz Ziv**
>
>
> -1. That the compression of the representation (reducing I(T:X)) is due to the saturated non-linearity and is not appear with other non-linearity (RelU’s in particular). The authors don’t know how to estimate mutual information correctly. When properly done, there essentially the same fitting and compression phases with RelU’s and any other network we examined: Here are the Information Plane trajectories for the CIFAR-10 CNN networks with RelU’s non-linearity as shown in our presentations: Figure 1 (see attachment). One can easily see the two phases and the phase transition between them (where I(X;T) has its maximum).
>
> In response to these concerns, we have run additional experiments and now include results using the state-of-the-art nonparametric KDE approach of Kolchinsky et al., (2017) as suggested in your prior comment, and the k-NN estimator of Kraskov et al, 2004. We find no compression with ReLUs or linear networks using (a) the exact MI estimation procedure used in “Opening the black box,” (b) the exact MI calculation with no approximations in linear networks, (c) the nonparametric KDE approach on the MNIST dataset, and (d) the Kraskov et al., (2004) k-NN based estimator. We note that, in addition to showing the robustness of our findings to the specific MI estimation method employed, these results also show the robustness to the particular dataset used: we find similar results using the original dataset of “Opening the….”, the linear student-teacher dataset, and now the MNIST dataset.
>
> -2. “that there is no evident causal connection between compression and generalization” We rigorously proved that compression leads to dramatic improvement in generalization, providing that the partitions remained homogenous to the label probability. In fact we argue that any bit of representation compression (under these conditions) is effective as doubling the size of the training data! Here is the sketch of our proof as given in our presentations: Figure 2 (see attachment).
>
> We note that the caveat is critical (“providing that the partitions remained homogenous to the label probability”): reducing to a discrete representation in which all inputs associated with the same discrete value have the same class label would be of benefit if possible. However no rigorous argument is given for why deep networks might achieve this special label-homogeneous partition if they in fact do, and this would seem to be the core fact to be explained. We also note that we observe similar generalization performance between Tanh and ReLU networks despite different compression dynamics, indicating that compression is not a major factor in the empirical behavior we observe (for instance, the bound might apply but be too weak).
>
> -3. “that the compression is unrelated to the noisy (low SNR) phase of the gradients”, as we claim. Below are some figures that clearly show the precise relation between the beginning of the compression phase (argmax I(X;T) for the last hidden layer (green line on left) and the gradient-SNR transition (blue line on right): Figure 3 (see attachment) Moreover, when changing the min-batch size (from 32 to 4000) both transitions move together in perfect linear relationship (left). In fact we show (right) that the full batch case (BGD of the “paper”) lies on the same line (green point) which suggests that the reported compression here is exactly the same phenomena, for much weaker gradient noise (as we claimed). We believe these facts nullify the arguments given in this “paper” all together.
>
> We have added plots of the gradient SNR to the appendix. As can be seen, in all cases (tanh, ReLU, and linear) we observe the two phases of gradient SNR (as is to be expected, e.g., Murata, 1998; Chee & Toulis, 2017). However we see no compression for ReLU or linear networks, indicating that these phenomena are unrelated. That is, because ReLU/linear networks do not compress, the plots given in the attachment would be vertical lines with no correlation between batch size and argrmax I(X;T) or between the gradient SNR phase transition and argmax I(X;T) (because argmax I(X;T) would be the final epoch of training in networks that do not compress). Hence, while we agree that this correlation exists for tanh networks, it does not for ReLU/linear networks, and therefore noise in the training process is not the causal mechanism behind compression. (We also note that there may be a plotting issue here—the x-axis of the left panel is labeled argmax I(X;T) and ranges from 0 to ~2000, while the y-axis of the right panel is also labeled argmax I(X;T) but ranges from 0 to ~200.)

---

### Author Response · Authors · 2018-01-04
**New revision posted**

We thank all reviewers for their thoughtful comments which have greatly improved the paper. We have just posted a revision which contains changes and additional experiments suggested by the reviewers. Most notably, we have replicated our basic results using the nonparametric KDE estimator (Kolchinsky et al., 2017) suggested for use in the OpenReview discussion, and using the popular k-NN based Kraskov et al., 2004 estimator. Again we find that ReLU networks do not compress while Tanh networks do. We also apply the KDE estimator to networks trained on MNIST, to show that the phenomenon also holds on (small) real world tasks. Additionally, we now include results for two other nonlinearities (soft plus and soft sign), where again only the double saturating nonlinearities show compression. We have also added results concerning the relationship between the two phases of gradient descent and compression: we show that networks exhibit these two phases regardless of the nonlinearity employed (for tanh, relu, or linear), and hence the SGD phases cannot be causally related to compression since networks that do and do not compress still exhibit them. We have endeavored to clarify points in the text, and now include extended discussions of several important points in the Appendix. We emphasize that all results reported in the original submission remain correct to our knowledge, and are left in place largely untouched—our additional results provide more thorough supporting evidence and control experiments that better establish the generality of our findings.

---

> ### Public Comment · ~Luis_Gonzalo_Sanchez_Giraldo1 · 2018-03-07
> **The argument based on the minimalistic model does not consider biases which make ReLUs behave nonmonotonically.**
>
> The minimalistic examples considered in the paper disregard the effects of the bias.
> If instead of having h = f(w*x), we use h = f(w*x + b), the mutual information I(X;T) does not behave monotonically.
> https://imgur.com/gO2y84h
> The color lines correspond to different biases. The binning uses 3000 bins uniformly distributed between 0 and 40.

---

### Public Comment · ~Jae_Duk_Seo1 · 2018-11-06
**Great work and thank you for opening this discussion.**

I am very happy to see these kinds of discussion happening, it seems like as a humanity we are moving in the right direction.

---

### Decision · Program_Chairs · 2018-01-29
**ICLR 2018 Conference Acceptance Decision**

**Decision:**

Accept (Poster)

**Comment:**


This submission explores recent theoretical work by Shwartz-Ziv and Tishby on explaining the generalization ability of deep networks. The paper gives counter-examples that suggest aspects of the theory might not be relevant for all neural networks.

There is some uncertainty surrounding the results where mutual information is estimated empirically. Even state-of-the-art estimation methods might lead to misleading empirical results. However, the submission appears to follow reasonable practice following previous work, making the reported results at least suggestive. They warrant reporting for further study and discussion.

The reviewers generally found the paper interesting enough for acceptance, however strong objections were posted by Tishby. A lengthy public exchange resulted between the groups of authors. Not every part of this exchange is resolved. It is not clear whether Tishby's group would be able to fix the full-connected ReLU demonstration in this paper, or whether the authors of this submission have anything to say about Tishby's ReLU+convnet demonstration. By accepting this work, we are not declaring where this debate will end. However, we felt the current submission is a constructive part of ongoing discussion in the literature on furthering our theoretical understanding of neural networks.